

# Sea ice data assimilation in ORAS6

Philip Browne, Eric de Boisseson, Sarah Keeley, Charles Pelletier, and Hao Zuo

European Centre for Medium-Range Weather Forecasts (Reading, United Kingdom. Bonn, Germany. Bologna, Italy)

**Correspondence:** Philip Browne (philip.browne@ecmwf.int)

**Abstract.** Accurate weather and climate forecasting relies heavily on the precise modeling of sea ice, a critical component of the Earth's climate system. Sea ice influences global weather patterns, ocean circulation, and the exchange of heat and moisture between the atmosphere and oceans. Initialisation of the sea ice component of global coupled models relies on data assimilation techniques to incorporate information from observations to constrain the system.

This study focuses on the development of sea ice data assimilation for ECMWF's latest Ocean Reanalysis System 6 (ORAS6) that includes a multicategory sea ice model. The research addresses the challenge of appropriately distributing sea ice concentration increments across various thickness categories in the model.

Here, we show that using a simple proportional increment splitting method improves the accuracy of sea ice concentration analyses compared to previous approaches. Our findings indicate that adding an additional sea ice-sea water temperature balance brings further performance benefits.

These results suggest that the choice of increment distribution strategy significantly impacts the accuracy of sea ice representation in reanalysis systems. The system presented here will form the basis of ECMWF's data assimilation system for numerical weather prediction, as well as the next generation coupled reanalyses.

## 1 Introduction

Numerical Weather Prediction (NWP) models serve as essential tools for forecasting atmospheric conditions, aiding decision-making processes across various sectors. The accuracy of these predictions critically hinges on the fidelity of initial conditions, particularly in regions with complex dynamics such as the polar oceans. Sea ice, a dynamic and sensitive component of these environments, plays a pivotal role in shaping climate and weather patterns by acting as a (thermo)dynamical isolating cover between the free ocean and the atmosphere. This paper explores the advancements in sea ice data assimilation within ECMWF's

latest Ocean Reanalysis System 6 (ORAS6).

Since 2018 ECMWF has had a dynamic sea ice model in all operational forecast systems (Keeley and Mogensen, 2018). It has been made possible by introducing the Louvain-la-Neuve Sea Ice Model (LIM2, see Fichefet and Maqueda (1997)) within ECMWF's ocean and sea-ice reanalysis system 5 (ORAS5, see Zuo et al. (2019)) and implementing assimilation of sea-ice concentration (SIC) data. Sea-ice assimilation in ORAS5 is conducted with NEMOVAR in its 3DVar-FGAT (First Guess at Ap-

propriate Time) configuration (Mogensen et al., 2012), by taking a univariate approach, meaning that neither cross-covariance between sea-ice state variables and ocean state variables nor covariance between sea-ice state variables (e.g between sea-ice



concentration and thickness) is accounted for. Many other methods have been successfully applied to sea ice assimilation in recent years. Notably Sakov et al. (2012) and Mu et al. (2020) applied Ensemble Kalman filter approaches, and Jean-Michel et al. (2021) has applied a Singular Evolutive Extended Kalman (SEEK) filter methodology to sea ice assimilation. Sea ice concentration is a two-sided bounded quantity, and as such many of the traditional variational assumptions of Gaussianity do not hold. Recent work from Cipollone et al. (2023) applies anamorphosis (transformations) to sea ice DA in order to account for such bounded probability density functions, and Wieringa et al. (2023) have carefully considered the impact of boundedness in the context of a range of Ensemble Kalman filter methods.

Other developments on sea-ice assimilation including attempt to constrain sea-ice thickness together with sea-ice concentration in the initial condition of coupled forecasts (Blockley and Peterson, 2018; Balan-Sarojini et al., 2020), with some promising results showing good impact for summer sea-ice forecasts on the seasonal scale.

In the years since the previous reanalysis at ECMWF was implemented, the sea ice model, which is part of the NEMO release, has changed from LIM2 to SI$^3$(Vancoppenolle et al., 2023) which has some important structural differences that have implications for the data assimilation system. Most crucial is the representation of the subgridscale distribution sea ice thickness, using an ice thickness distribution for which sea ice concentration values need to be initialised. It is no longer sufficient for a single value at a grid point to be generated as an analysis product. Besides the thickness distribution, the updated sea-ice model also features significantly more sophisticated physics. The challenge for data assimilation is how best to incorporate observational information of sea ice concentration to initialise and constrain this modern configuration of model.

This paper will discuss the technical developments and scientific choices we have made when developing sea ice data assimilation in the new multicategory sea ice model SI$^3$. We will present the model and final settings used for ORAS6, before justifying these by showing 5 year long experiments comparing other possible configurations. We conclude with some thoughts on future developments and other observational sources which will help constrain the new model's representation of the ice state.

## 2 SI$^3$ model and multicategory representation of sea ice

There are many developments in the SI$^3$ compared to LIM2, here the differences pertinent to the data assimilation system are highlighted; the largest difference is the way that the heterogeneity of the sea ice is modelled in LIM2 and SI$^3$. For SI$^3$, the model employs an Ice Thickness Distribution (ITD) scheme [Thorndike et al. (1975),Bitz et al. (2001), see also our Figure 1], allowing it to represent subgridscale variations in important processes such as energy exchanges that depend nonlinearly on the thickness of sea-ice. This scheme provides a detailed calculation of the energy balance at the ice surface, considering ice properties, such as albedo and enthalpy, to be specific to each ice category, which is discretized into four vertical layers with a single layer of snow on top. In contrast, LIM2 does not use an ITD scheme, this is parameterized and it represents sea ice with a single ice thickness category, with two vertical layers with a single snow layer, leading to a less detailed portrayal of the associated processes. Additionally SI$^3$ represents the thermodynamic properties of the sea ice differently to LIM2, allowing salinity to vary rather than assuming a fixed ice salinity and uses enthalpy rather than temperature, as in LIM2, as a





prognostic variable. SI$^3$ also benefits from several developments to parametrisation schemes beyond those employed in LIM2, most significantly the formation and evolution of melt ponds. There are also developments in the parameterizations for the albedo of snow-covered, ponded and bare ice surfaces, which dynamically change based on surface conditions. Additionally, the snow scheme also includes a blowing snow parameterization. SI$^3$ uses an Elastic-Viscous-Plastic (EVP) rheology (Bouillon et al., 2009), whereas in contrast, LIM2 employs a Viscous-Plastic (VP) rheology. SI$^3$'s EVP model is a relaxed form of the

equations which are able to be solved in a more computationally efficient way than the VP scheme of LIM2.

Overall, SI$^3$ provides a more detailed and accurate representation of sea ice and snow processes compared to LIM2, which results in a more uniform and less complex depiction of these processes. This difference can significantly impact the accuracy of energy balance calculations and the simulation of ice evolution in different environmental conditions.

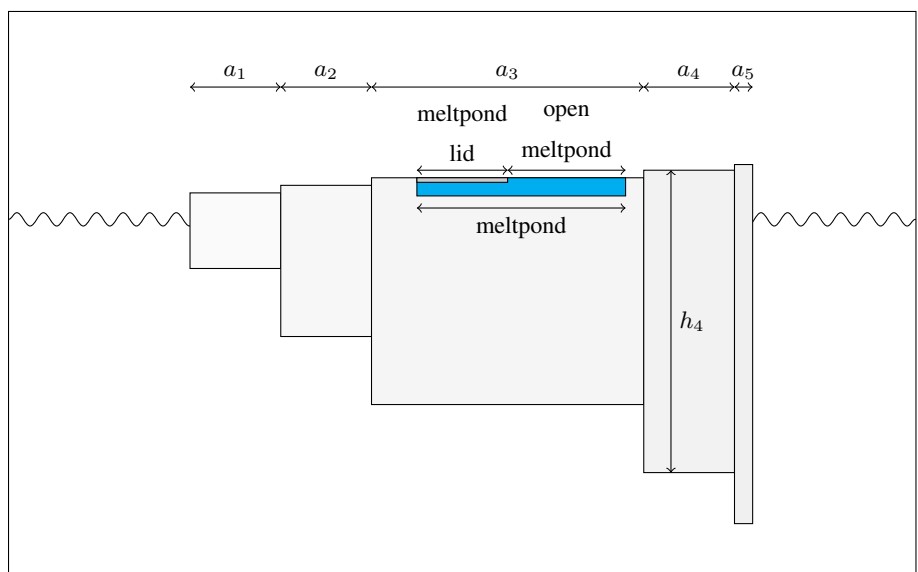

**Figure 1.** Schematic of SI$^3$ multicategory approach. There are $N$ different ice thickness categories alongside open water in the non-ice covered region of the gridbox. Each ice thickness category has many prognostic variables, for instance melt ponds with lids as shown. Not shown are vertical layers used to have a temperature profile, as well as snow in each category.

For ORAS6 SI$^3$ is configured with different ice thickness categories $j$ from 1, the thinnest to $N = 5$, the thickest ice. The

thickness category boundaries implemented are 0.45m, 1.13m, 2.14m, and 3.67m. This means that there is an ice concentration associated with ice with thickness somewhere between 0m and 0.45m, another ice concentration associated with ice of thickness somewhere between 0.45m and 1.13m, and so on.

## 3  Sea ice concentration observations

A number of observation operators have been developed for sea ice observations within SI$^3$. All of the below equations repre-

sent a single grid point, and so are computed for each spatial location. The most basic is the grid point total sea ice concentration





(SIC), $a$, where

$$a := \mathcal{H}_{\mathrm{conc}}(x) = \sum_{j=1}^{N} a_j. \tag{1}$$

This total concentration across the categories is equivalent to the average SIC in the grid box. There is an optional feature to include the effect of melt ponds on detectable sea ice concentration:

$$\mathcal{H}_{\mathrm{conc\ mp}}(x) = \sum_{j=1}^{N} a_j(1 - p_j) \tag{2}$$

where $p_j$ is the *effective* melt pond fraction in category $j$.

For ORAS6 we are only actively assimilating sea ice concentration observations using the observation operator given by (1), and we assume that the products we are using have already been corrected for the presence of melt ponds.

For production of ORAS6 we have to choose different observations depending on the period. Table 1 summarises these
choices, the main being Ocean and Sea Ice Satellite Application Framework (OSI SAF) data in the satellite era. For the pre-satellite period, the choice of observation data source and assimilation configuration has not yet been finalised.

| Start date | End Date | Product |
|---|---|---|
| 1972-12-11 | 1977-05-11 | ESA SIC CCI Nimbus-5 |
| 1978-10-25 | 2020-12-31 | OSI-450-a |
| 2021-01-01 | 2025-03-15 | OSI-430-a |
| 2025-03-16 | 2025-07-05 | OSI-430-a fast track |
| 2025-07-06 | - | OSI-438 fast track |

**Table 1.** Input sea ice concentration observational products by date

### 3.1 Level 3 sea ice concentration observations from OSISAF

OSI-450-a is the third version of a reprocessing of sea ice concentration based on SMMR (1978-1987), SSM/I (1987-2008) and SSMIS (2006-2020). Atmospheric data from ERA5 is used in its production. It is a daily level 3 product covering the
northern and southern hemispheres. OSI-430-a is an extension of the same processing chain as OSI-450-a, but with input data from SSMIS and ERA5-T atmospheric fields. OSI-430-a has a latency of 16 days, which is not suitable for the continuous production schedule of ORAS6, in which case the fast track version is used as it has a suitable latency of 2 days. The fast track product uses atmospheric fields from the ECMWF operational analysis rather than ERA5 or ERA5-T. From July 2025 onwards, ORAS6 is ingesting an AMSR2 based SIC product, OSI-438. All of these OSISAF products have a nominal resolution of 25km.
Recent data rescue efforts have led to SIC products being generated from NIMBUS-5 in the 1970s (Kolbe et al., 2024; Tonboe et al., 2025). These products, generated by the ESA Climate Change Initiative, are produced on the same grid as the above OSISAF products. Whilst they are slightly noisier, more prone to gaps, and have more patches of missing data than





the OSISAF products in more established satellite periods, they still are of sufficient quality to enhance the reanalysis. As the methodology is assuming L3 input, the gaps and missing data are not a concern for the assimilation system.

The data is randomly thinned to boxes of 0.25/0.25 degrees. Perturbed members of the ORAS6 EDA assimilate perturbed SIC observations. The perturbations are generated by sampling from a database of both analysis and structural errors. The analysis error database is constructed from the differences between ERA5 ensemble members, and the structural error database is constructed from the differences between OSTIA (Good et al., 2020) and ESA SST CCI v2 Merchant et al. (2019) observational datasets.

## 4    Multicategory sea ice assimilation methodology

We have seen that in SI$^3$ the model state is split across multiple sea ice concentration categories. However we have not currently changed the data assimilation system to work natively with each category of ice concentration. Instead, the variational minimisation retains a single category control vector and the challenge is then to consistently apply a single increment to each of the categories.

### 4.1    Computation of single category increment (NEMOVAR settings)

Unlike the ocean component of ORAS6, the variational scheme used retained many of the settings that were in use in ORAS5. The background error covariance matrix is based on a diffusion operator with a horizontal length scale of 2 degrees, with background error standard deviations a constant 0.05. The observation errors used are also a constant 0.2.

Sea ice concentration is included in a joint minimisation with the ocean, unlike in ORAS5 where L4 sea ice concentration
observations were assimilated separately. This means that sea ice concentration is part of the NEMOVAR control vector along-side 3D temperature, salinity, $u/v$ currents, and sea surface height. However there remains no terms in the background error covariance or the NEMOVAR balance operator that couple the sea ice to the ocean state. This was possible as we found in the ORAS6 system, with its revised background error covariance configuration, joint minimisation gave neutral scores compared to separate minimisations. Use of L3 sea ice concentration observations may be playing a role in allowing the joint minimisation
as fewer SIC observations could be allowing the minimisation to successfully converge in both ocean and ice states.

### 4.2    Distribution of single category increment into multicategory model

Given an increment that is one value valid within a grid-box of sea ice concentration, how does one choose to assign this to the different thickness categories which make up the total ice within the grid-box?

This is not a well defined problem. For example you could choose to add all increments to the thinnest category, or you could
fit a fixed distribution to the increments such as a gamma profile. These were indeed trialled but ultimately not used (see results in section 6.1). Noting that sea ice concentration is orthogonal to sea ice thickness, it is desirable that sea ice concentration increments should not change the sea ice thickness.



The strategy we adopt so that ice thickness is not affected by concentration increments is to distribute the increment proportionally to the existing model distribution of ice thickness. For a given ice increment $\delta a$, we compute the increment for each
category, $\delta a_j$ as

$$\delta a_j = \frac{a_j}{\sum_{j=1}^{N} a_j} \delta a. \tag{3}$$

In effect means that the increment is distributed in proportion to the existing ice thickness distribution.

So if $h^{\text{old}}$ is the thickness of the existing ice, then

$$h^{\text{old}} = \frac{\sum_{j=1}^{N} a_j h_j}{\sum_{j=1}^{N} a_j}.$$

The new ice thickness is

$$h^{\text{new}} = \frac{\sum_{j=1}^{N} (a_j + \delta a_j) h_j}{\sum_{j=1}^{N} (a_j + \delta a_j)} = \frac{\sum_{j=1}^{N} (a_j + \frac{a_j}{a} \delta a) h_j}{\sum_{j=1}^{N} (a_j + \frac{a_j}{a} \delta a)} = \frac{(1 + \frac{\delta a}{a}) \sum_{j=1}^{N} a_j h_j}{(1 + \frac{\delta a}{a}) \sum_{j=1}^{N} a_j} = h^{\text{old}}.$$

Thus this choice of splitting the increment across the different thickness categories does not impact the sea ice thickness. This methodology is only applicable when there is some existing sea ice concentration within the grid box; otherwise there will be a division by 0. Section 4.3.2 covers the special case of adding an increment when no ice exists in the model.

## 4.3   Application of increments

Many sea ice variables are inherently bounded and therefore do not follow the common assumption that errors have a Gaussian distribution. For example, thickness is bounded below by 0, and concentration is bounded below by 0 and above by a constant $\leq 1$.

None of these constraints are applied within the minimisation process of the analysis. Instead, the constraints are respected
when the increment is applied to the model. The net result of this is that the increments are effectively truncated if they were to take the model state out of bounds.

### 4.3.1   To existing ice pack

Many SI$^3$ prognostic variables are extensive, meaning that they depend on the size of the system: in our case, the size is the sea ice or snow volume per unit area, which is category-specific. Physically intensive quantities (e.g., ice thickness, ice
temperature, melt pond thickness, ice salinity) can then be diagnosed from their extensive counterparts (e.g., respectively ice volume, ice enthalpy, melt pond volume, ice salinity-volume product).

When modifying sea ice concentration we want to ensure that all extensive quantities are updated to reflect the changes in volume, due to the concentration changes. This ensures the conservation of intensive quantities throughout the assimilation. For example, updating the ice enthalpy is needed for preserving the ice at the same temperature as before the assimilation
increment.





So for some extensive quantity $\gamma$, its value after applying a concentration increment $\delta a_j$ is

$$\gamma^{\text{new}} = \gamma^{\text{old}} \frac{a_j + \delta a_j}{a_j}. \tag{4}$$

Equation (4) is applied where $\gamma$ is ice volume, ice enthalpy in each vertical layer, snow volume, snow enthalpy in each vertical layer, the salinity volume product, and meltpond concentrations/volumes/lid volumes. Equation (4) is not applied to the sea ice areal age content as this could not be made numerically stable, potentially because sea ice areal age is proportional to concentration, not volume, which is an exception in SI³. Moreover, sea ice areal age is a diagnostic-only variable; not updating it therefore leaves the model trajectory unchanged (but decreases the relevancy of the sea ice age diagnostic). Across the 5 thickness categories, 4 thermodynamic ice layers, and single thermodynamic snow layer, this amounts to 55 prognostic fields that need to be updated in line with the increment to concentration.

### 4.3.2   To open water

Any negative ice increments where there is currently open water are disregarded. This leaves the situation where ice has to be added to open water, and there some choices need to be made. The thickness of the new ice needs to be specified. We choose 0.45m of ice thickness so that it remains entirely within the thinnest ice category (for the specific ice category boundaries we have). This is very similar to the 0.5m ice thickness used in ORAS5. The other sea ice properties are set so that the sea ice created by the DA is identical to sea ice that would have thermodynamically formed into the open water. The salinity of the new ice thus uses the same settings as new ice that the model would form from open water, in our case a varying salinity parameterisation of Vancoppenolle et al. (2005). The new ice enthalpy is set so that it is at the freezing point temperature. In the absence of any other information the snow, meltponds, and the age of this new ice are all set to zero.

### 4.3.3   Ice induced temperature increments

Through the physical processes present in the NEMO/SI³ coupled system, sea ice will only be sustained when, for example, the sea water is cold enough that it will not immediately melt the ice. This is an example of a balance which would typically be approximated within the data assimilation analysis process. In the absence of a full careful implementation of such a balance within NEMOVAR we apply this as a simple post-processing of the output.

In simple terms, where there is a positive ice increment we seek to reduce the sea water temperature, and vice versa. However we have to ensure the relationship does not work in the other direction: a negative sea water temperature increment should not induce a positive sea ice increment, else for example sea ice could be added in tropical regions.

The effect of this ice induced temperature increment is to ensure the ice increment remains in the system and changes the ice state appropriately. It is not immediately counteracted by a potentially inconsistent sea water state which would not support the desired ice state.

The sea water temperature increment $\delta t$ in the vertical water column $z$ is modified in the following way

$$\delta t'(z) = \delta t(z) - \alpha f(z)\delta a, \tag{5}$$





where $\delta t'$ is the updated sea water temperature increment. $\alpha$ is a positive scalar controlling the magnitude, and $f(z)$ is a normalised vertical profile controlling how deep into the water column the induced increment propagates. We implement a very basic $f(z)$ where it is 1 at the surface and decays linearly to a minimum of 0 at the 12th model level (approximately 19.5m).

### 4.3.4 Interaction of increments with high order conservative advection scheme

SI[3] has a few different advection schemes available, but the one chosen for our configuration was the scheme of Prather (1986). This scheme is designed so that the advection will conserve not only a tracer, but also the first and second order spatial gradients of the tracer. The interaction of this scheme with the addition of data assimilation increments through the Incremental Analysis Update (IAU) can cause problems.

In the IAU process, the increment is added not all at once at the beginning of a model run, but partially at every model timestep. This is beneficial as it reduces initialisation shock and allows the model to dynamically adjust to the new state. This IAU process is by definition non-conservative - some of the 0-order quantities which undergo advection are modified every timestep.

If the values of the tracer itself are changing, then conserving first and second order gradients does not make sense, as those gradients would not be representative of the updated tracers. We found this inconsistency first appearing in the ice salinity fields, where numerical chequerboard patterns appear as in Figure 2. A particular feature of this application is that within the sea ice pack there is almost no diffusion of the salinity spatially. Once the spurious patterns form they remain in the model until the ice pack melts entirely at the effected locations. Such noisy salinity fields affected the surface temperatures and therefore fed through to degraded coupled model performance.

To mitigate this, we modify the advection scheme so that when IAU is active the spatial gradients are recomputed every model timestep. This ensures the higher order moments are consistent with the new model state, and effectively removes the chequerboard patterns as seen in Figure 2a.

## 5 Experimental framework

The experiments which we perform are all directly comparable with the production ORAS6 reanalysis. To briefly summarize the ORAS6 system, it uses NEMO v4.0 and the SI[3] sea ice models on the eORCA025 tripolar horizontal grid, which is approximately 27km at the equator and 12km in the Arctic (Bernard et al., 2006). Hourly data from ERAS5 provides the atmospheric forcing. In situ profiles from ARGO, moorings, ships, mammals, and gliders are assimilated alongside satellite altimeter data, L4 SST data, and the previously described sea ice concentration data.

The assimilation system used is NEMOVAR in 3D-Var FGAT mode with a 5 day assimilation window. An Ensemble of Data Assimilations (EDA) is used consisting of one control member is used, alongside 10 perturbed members (perturbed observations and forcing). The EDA provides a flow dependent component to the background error variances used in the



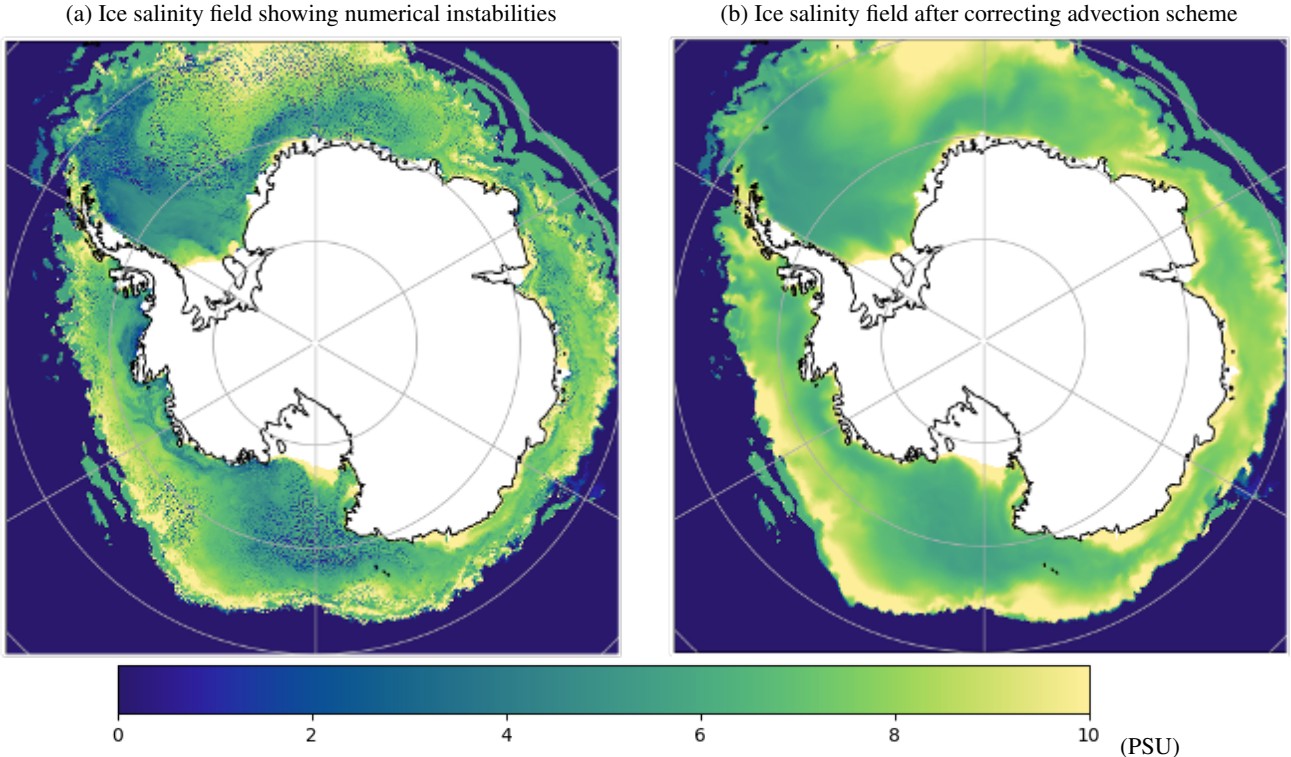

**Figure 2.** Effect of (not) accounting for the data assimilation increment in the advection scheme used with SI[3]. The numerical noise seen in the left hand plot persists due to the lack of diffusion of salinity within the ice pack.

ocean, as well as flow dependent vertical correlation structures. For an overall description of the system, please see Zuo et al. (2024) and for details of the EDA please see Chrust et al. (2024).

220    In this paper we show results from deterministic (single member) experiments, using the ORAS6 EDA. Experiments all start from the ORAS6 control on 2010-01-01 and run for a period of 5 years.

We perform 3 sets of experiments: the first to assess the impact of distribution strategies of the sea ice increment (see Section 4.2), second to assess the choice of how to add ice to open water (see Section 4.3.2), and finally experiments to assess the impact of inducing temperature increments from the sea ice concentration increment (see Section 4.3.3).



## 6 Experiments and results

### 6.1 Single category to multicategory increment distribution

The control, ORAS6, splits its increment by following (3) so that it remains in proportion to the background profile of sea ice concentration across the multiple ice thickness categories at each gridpoint. We run two further experiment with different splitting strategies.

The first where we use the scheme of Peterson et al. (2015) that adds positive increments to the thinnest ice category, and negative increments to the thinnest available category and if it reaches zero concentration then progressively removes ice from the next thickest categories.

The other experiment assigns a gamma profile with scale factor 2 to the magnitude of the increments. This follows Abraham et al. (2015), where the pdf of the thickness distribution in terms of $h$ can be expressed in terms of the grid-point mean thickness, $\bar{h}$, as follows.

$$\text{pdf}(h) = \frac{4h}{\bar{h}^2} \exp\left(-\frac{2h}{\bar{h}}\right).$$

Illustrative density functions following a gamma distribution are shown in Figure 3 for different ice thickness values. To get the increment per category we integrate this function between the lower and upper limits of thickness in each category and then scale by total concentration increment.

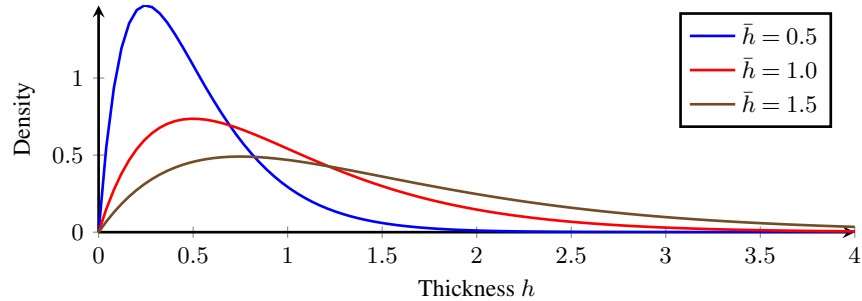

**Figure 3.** Representative sea ice thickness distributions that follow a gamma distribution for various different mean sea ice thicknesses.

The scores shown in Table 2 show that both the gamma splitting and the background splitting of the increment achieve significant improvements over the Peterson splitting. The Peterson splitting by its nature will thicken the ice pack, as where ice concentration values are high, increments are more likely to be negative than positive. This means that this splitting strategy will be removing the thinnest ice in those regions, thus increasing the grid point average thickness. Whilst this does have some benefit, we chose not to use the increment splitting mechanism to address what are model/forcing biases best corrected at source.




|  | Global (err) |  | N. Hemisphere (err) |  | S. Hemisphere (err) |  |
| --- | --- | --- | --- | --- | --- | --- |
| Background splitting | 100 | - | 100 | - | 100 | - |
| Peterson splitting | 107.086 | 0.671375 | 107.230 | 1.0463 | 106.074 | 1.03685 |
| Gamma splitting | 99.2528 | 0.07462 | 99.1829 | 0.130005 | 99.3014 | 0.101979 |

**Table 2.** SIC o-b standard deviations, normalised by the standard deviation of the control. The second columns show the magnitude of a 95% confidence interval. Using the splitting mechanism of Peterson et al. (2015) leads to around 7% degradation in background fits to SIC observations, whereas using a Gamma splitting leads to a 0.75% improvement.

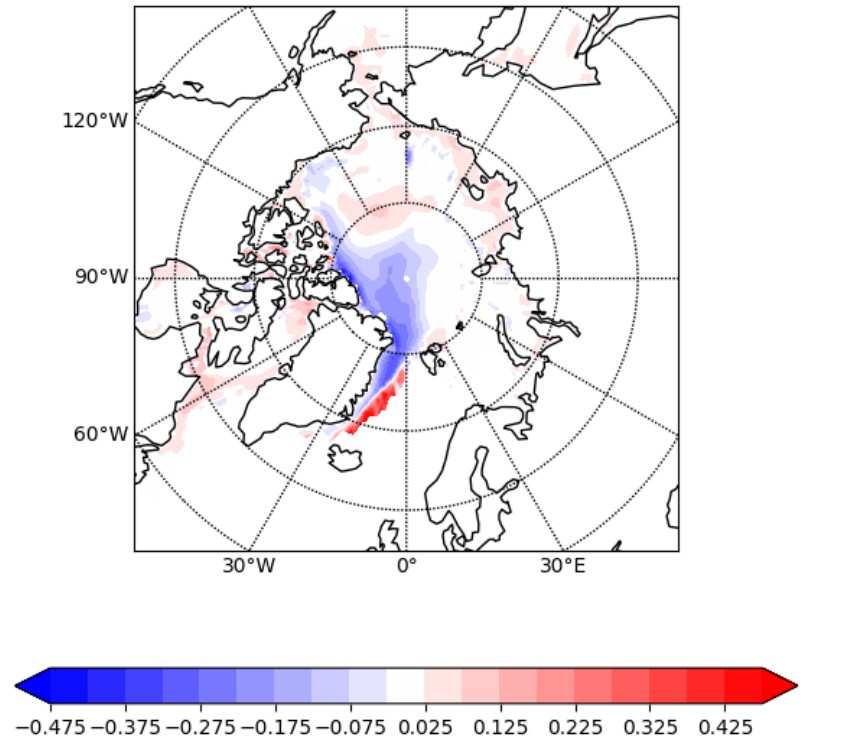

**Figure 4.** Errors in analysis of sea ice thickness comparing to *CS2SMOS merged CryoSat-2 and SMOS ice thickness*. Shown is the difference in RMSE using the gamma splitting compared to background splitting. In red are regions of worse sea ice thickness, which is notable around the ice edge in the Atlantic.

The gamma splitting mechanism does give the best results in terms of concentration, but it has worse thickness performance around the ice edge in the Atlantic sector (Figure 4). The ECMWF NWP system is sensitive to sea ice in this region, and so whilst SIC performance was the best we have chosen to implement the background profile splitting.



## 6.2 Addition of ice to open water

The control, ORAS6, sets the thickness of new ice to 0.45m, the maximum thickness of the thinnest sea ice category. We run two further experiments, one where we add new ice with a thickness of 0.1m, and another with thickness of 0.225m.

|  | Global (err) | | N. Hemisphere (err) | | S. Hemisphere (err) | |
|---|---|---|---|---|---|---|
| 45cm | 100 | - | 100 | - | 100 | - |
| 22.5cm | 100.631 | 0.0576538 | 100.333 | 0.0582571 | 100.924 | 0.0938093 |
| 10cm | 101.182 | 0.0825902 | 100.489 | 0.0648830 | 101.824 | 0.144142 |

**Table 3.** SIC o-b standard deviations, normalised by the standard deviation of the control. The second columns show the magnitude of a 95% confidence interval. Adding ice to open water with only 10cm thickness degrades the o-b fits by up to 1.8% (southern hemisphere) and 22.5cm thick ice degrades by around 0.6% (global).

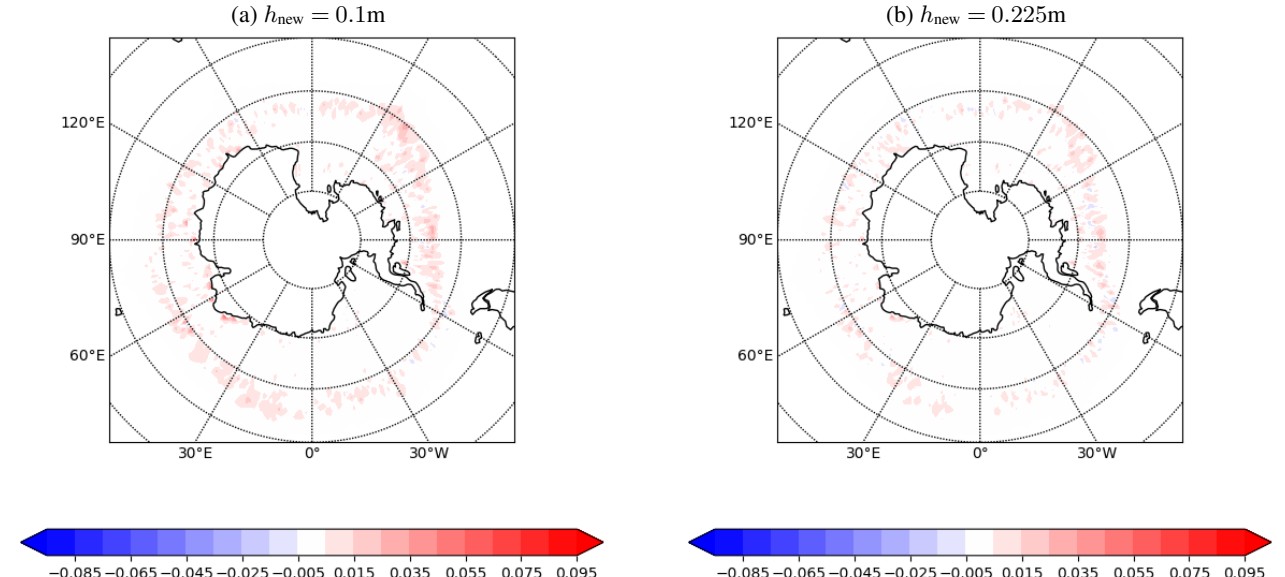

**Figure 5.** Southern hemisphere SIC o-b RMSE differences compared to using a thickness of new sea ice $h_{new} = 0.45$m. Larger errors are seen around the ice edge when thinner ice is assumed to be missing at open water points.

Table 3 shows the results from assuming different thicknesses of new ice formed by the data assimilation. It shows that adding ice at the upper threshold of the thinnest ice category gives the best performance. Figure 5 highlights that the impact is confined to the ice edge as expected.




## 6.3 Sea ice–water temperature balance

The control, ORAS6, implements a basic postprocessing balance relationship between the sea ice increment and the sea water temperature by following (5). For brevity, we will keep the structure of the vertical profile, $f(z)$, fixed and experiment with the magnitude of the induced increment. ORAS6 uses $\alpha = 5$ to define this magnitude. We run two further experiments with $\alpha = 0$ and $\alpha = 2.5$ to assess the impact of this choice.

In a simple linear system, the analysis $x_a$ should be given by the addition of the increment to the background, $x_b + \delta x$. If $x_a - x_b \not\approx \delta x$, this shows the increment is not having the desired effect on the analysis. Figures 6 and 7 show how, in a single cycle, the inclusion of the ice induced temperature increment allows the effective increment (analysis-background) to better correspond to the minimisation increment.

|  | Global (err) | | N. Hemisphere (err) | | S. Hemisphere (err) | |
|---|---|---|---|---|---|---|
| $\alpha = 5$ | 100 | - | 100 | - | 100 | - |
| $\alpha = 2.5$ | 102.466 | 0.199485 | 101.746 | 0.319479 | 102.793 | 0.240604 |
| $\alpha = 0$ | 107.665 | 0.324827 | 106.443 | 0.487622 | 108.399 | 0.424248 |

**Table 4.** SIC o-b standard deviations, normalised by the standard deviation of the control. The second columns show the magnitude of a 95% confidence interval. Having no ice induced temperature increment degrades o-b fits by 7.7% globally, and reducing the magnitude by half degrades the fit by 2.5% globally.

|  | Global (err) | | N. Hemisphere (err) | | S. Hemisphere (err) | |
|---|---|---|---|---|---|---|
| $\alpha = 5$ | 100 | - | 100 | - | 100 | - |
| $\alpha = 2.5$ | 104.938 | 0.263607 | 104.236 | 0.360889 | 105.119 | 0.298413 |
| $\alpha = 0$ | 114.485 | 0.510115 | 114.024 | 0.796284 | 114.386 | 0.642265 |

**Table 5.** As Table 4 but for o-a (analysis) fits to SIC. Having no ice induced temperature increment degrades o-a fit by over 14% globally.

Tables 4 and 5 give details of the performance of different magnitudes of the ice induced temperature increments ($\alpha$ in equation (5)). Figure 8 shows, over the full 5 year test period, that the ice induced temperature increment has a consistently positive impact not just on the ice edge, but also throughout the ice pack.





(a) Increment of sea ice concentration

(b) analysis - background for $\alpha = 5$

(c) analysis - background for $\alpha = 2.5$

(d) analysis - background for $\alpha = 0$

**Figure 6.** Northern Hemisphere sea ice concentration increment and effective increments for various strengths of the ice induced temperature increment for the first analysis cycle 2010-01-01:2010-01-05



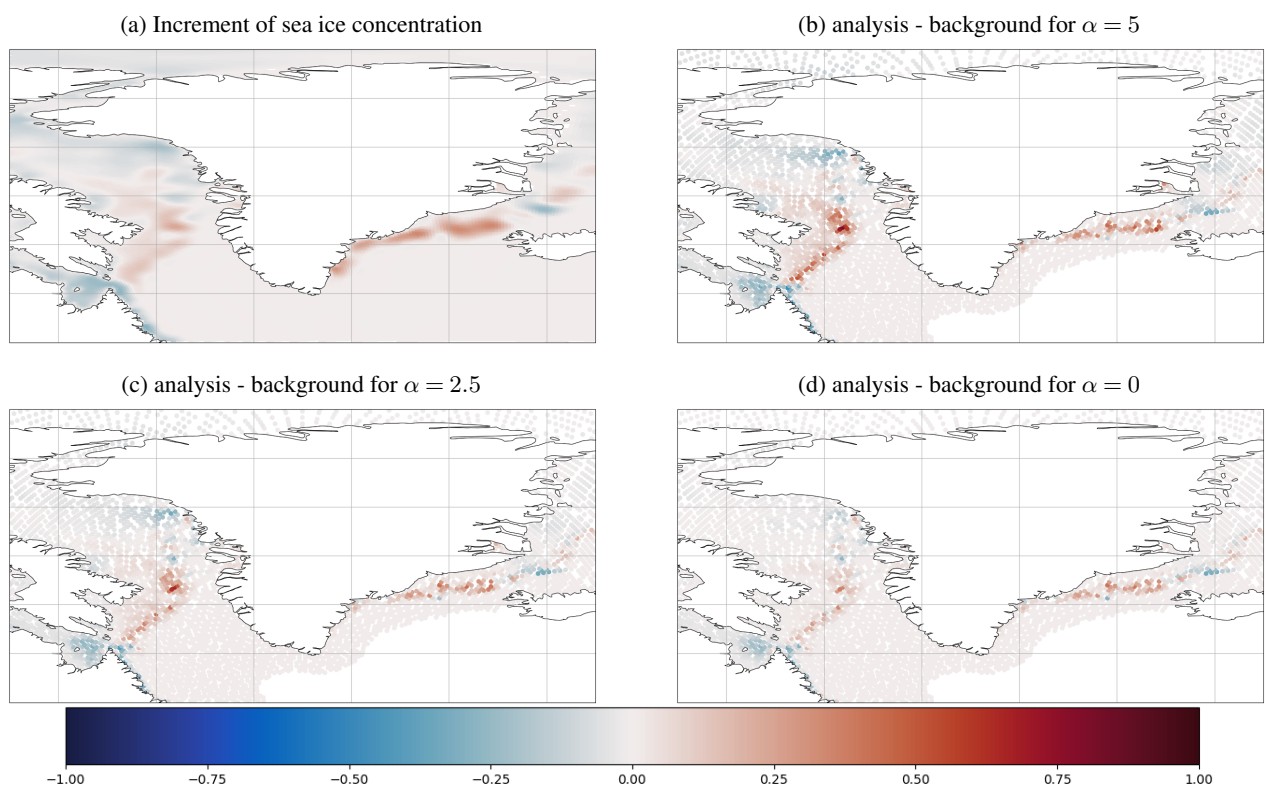

**Figure 7.** Greenland and Labrador sea sea ice concentration increment and effective increments for various strengths of the ice induced temperature increment for the first analysis cycle 2010-01-01:2010-01-05





**Figure 8.** SIC o-b RMSE differences compared to using a magnitude of the ice induced temperature increment given by $\alpha = 5$.



## 7 Conclusions and planned developments

Sea ice concentration is an important dataset to be assimilated into an ocean reanalysis. We have detailed the developments we have made and the configuration used to assimilate sea ice concentration into the multicategory sea ice model SI$^3$ within the ORAS6 reanalysis.

Ice induced temperature increments give the largest impact of all developments.

We have shown that results are highly sensitive to the method used to distribute a single category increment across the multiple prognostic thickness categories of the model. Our choice to keep the increments in proportion to the background profile of ice distribution was shown to perform well, and has the benefit of leaving the sea ice thickness unaffected, by design.

Future developments for the assimilation into a multicategory sea ice model will focus on including the sea ice concentration of each thickness category directly within the data assimilation control vector. In doing so there is sensitivity to sea ice thickness observations which would allow them to be assimilated. Inter-category relationships/constraints will have to be managed either through the background error covariance or a balance operator, and the best way to do this will be investigated.

Sea ice thickness observations are currently not used in ORAS6, but previous studies at ECMWF (Balan-Sarojini et al., 2020) and elsewhere (Mignac et al., 2022) have shown their potential for improving both the analysis and NWP. However only a relatively short timeseries of suitable data is available (from 2010 onwards from the Cryosat-2/SMOS missions) in the northern hemisphere (snow loading on the Antarctic sea ice is a challenge for retrievals) with gaps due to the presence of summer melt ponds. To use this data in an ocean/sea ice reanalysis smoothly without introducing artificial climate signals will be a challenge. Further the methodology must also be applicable to real time analyses so that NWP and seasonal forecasts can be initialised consistently to reforecasts. Use of datasets that have used machine learning methods to fill the traditional gaps (such as Landy et al. (2022)) would require significant methodological developments to first estimate appropriate error covariances before they could be assimilated. However if sea ice thickness information can be used to its full extent, there is a possibility that the ill-posed problem of distribution of increments across ice thickness categories could become locally well-posed.

The methodology presented here will be used as a baseline for the real time sea ice analysis at ECMWF. Work to optimise the background error covariances settings may be undertaken, in particular to constrain smaller scales, potentially using a hybrid EDA formulation to account for flow dependent errors.

In the near future multicategory sea ice assimilation will be part of the ECMWF coupled NWP system (de Rosnay et al., 2022). In that case the L3 observations assimilated here will be replaced by coupled brightness temperature retrievals (Geer, 2024), hopefully improving the system by improving the timeliness and reliability of the system as a whole.

*Data availability.* The ORAS6 reanalysis will be publically available via the ECMWF data dissemination services https://www.ecmwf.int/en/forecasts/datasets/browse-reanalysis-datasets.



*Author contributions.* PB designed the methodology within the wider methodology of ORAS6 led by HZ. PB, SK, CP and HZ contributed to the implementation. PB, EB and HZ performed experimentation. All authors contributed to the writing and revisions of the manuscript.

*Competing interests.* The authors declare that they have no conflict of interest.



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
