# Peer review of "Sea ice data assimilation in ORAS6"

_EGUsphere, 2025_

## Referee Comment (RC1)

This manuscript presents the development and evaluation of sea ice concentration (SIC) data assimilation within ECMWF's new Ocean Reanalysis System 6 (ORAS6), which incorporates the multicategory SI3 sea ice model. The authors describe modifications to the NEMOVAR data assimilation framework, including the distribution of SIC increments across thickness categories and an imposed ice—water temperature balance adjustment. The experiments are well-structured and represent a meaningful advancement toward more physically consistent SIC assimilation in a multicategory model.

That said, the paper requires substantial clarification and justification across multiple sections before it is ready for publication. Many of the presented choices (e.g., increment-splitting schemes, fixed new-ice thickness, and temperature adjustments) lack sufficient physical motivation or citation support. In addition, key assumptions—such as the claimed orthogonality between sea ice concentration and thickness—are questionable and should be more carefully examined or revised. Figures also require clearer labeling, scaling, and physical interpretation. Overall, the manuscript is promising but would benefit from a more transparent exposition of methodological decisions, improved figure presentation, and tighter linkage between the experimental setup and the stated objectives.

**Major Comments**

1. Conceptual and Methodological Clarity

**Lines 15-40+:**

Please define all acronyms (ECMWF, ORAS6, LIM2, SI3, NEMOVAR, etc.) upon first mention. The introduction assumes strong familiarity with ECMWF's system architecture, which may limit accessibility for broader readership.

**Lines 120–135:**

The statement in Line 126 that "sea ice concentration is orthogonal to sea ice thickness" requires careful revision. These quantities are not strictly orthogonal in the linear algebraic sense; rather, they are inherently linked through the ice volume relationship  $SIV_n = SIC_n \times SIT_n$ . For instance, during the melt season, increased atmospheric and oceanic temperatures cause thinner ice to melt more rapidly, leading to a decrease in SIC. Meanwhile, thicker ice (i.e., higher SIT) tends to persist longer, resulting in spatial patterns where SIC and SIT covary—contradicting the claim of orthogonality. During the freeze-up period, SIC and SIT may indeed covary less, but as illustrated above, this relationship is not universally negligible. Assuming zero covariance between the two may therefore misrepresent coupled ice processes and potentially lead to unphysical SIT (and consequently SIC) states. While SIC and SIT are often treated as distinct state variables, it is inaccurate to describe them as orthogonal. This assumption should be revisited, and the implications for the chosen increment distribution scheme should be discussed.

If SIC is updated independently across categories primarily for practical reasons, that rationale should be made explicit.

Following up on this point (and similar to the one above), since sea ice volume is an extensive variable and is updated as such in Equation 4 (Line 158), have you considered how restricting changes in the ITD impacts your updates? For example, imagine an update in the Central Arctic, where ice is relatively thick (>1 m). Suppose a ridging event occurs that leaves open water, and the ice then refreezes according to observations and the prevailing cold atmospheric state, warranting a positive SIC increment. This newly formed ice is likely thin to start, according to well-established theory, but under your current update scheme, it would be constrained to adhere to the original thickness distribution—potentially overcompensating and producing too much ice volume, leading to an unphysical amount of ice. Have you considered this or similar scenarios? The ice volume increment could be scaled more strongly toward thinner ice categories in both positive and negative increments to better represent the persistence of thicker ice, as widely documented in the literature.

**Lines 165–175:**

The approach of assigning a fixed thickness of 45 cm to newly formed ice is insufficiently justified. The sensitivity tests in Section 6.2, which explore the additional fixed thicknesses of 22.5 cm and 10 cm, are valuable but not comprehensive. A more physically consistent approach would be to assign thickness proportionally to the magnitude of the SIC increment (e.g., thinner ice—approximately 10 cm—for small increments and thicker ice—up to 45 cm—for larger increments). Please also provide the specific reference supporting the 0.5 m threshold used in ORAS5.

**Lines 180-195:**

The implementation of an "ice-induced temperature increment" is a creative solution to maintain thermodynamic balance, but its physical basis and vertical extent (to 19.5 m) are questionable. Please provide justification for the chosen  $\alpha$  values and the depth profile f(z), as decreasing ocean temperature at such depth could unintentionally alter stratification or vertical mixing. It is unclear whether this adjustment differentiates between increments arising from advection versus thermodynamics—this distinction should be addressed. For example, in the case of advection, atmospheric forcing from cyclones may move ice equatorward, which most likely does not alter the near-surface ocean profile in the same way that the thermodynamic growth of ice would.

**Lines 220-270:**

The experimental design is clearly described; however, several key methodological details are missing and appear to rely on the reader's prior familiarity with ORAS6. Please specify the number of ensemble members at the first mention (e.g., near Line 100, rather than only at Line 216). Further clarification is also needed regarding the "deterministic (single-member) experiments" described at Line 220—specifically, how they were implemented, why this approach was chosen, and whether any statistical significance testing was performed to account

for potential control-member dependence (e.g., rerunning with five different control members to assess robustness). Additionally, the presentation of normalized standard deviations in Tables 2–5 would be clearer if accompanied by absolute RMSE values and the sample size used for each experiment (e.g., n = ...).

2. Physical Justification and Link to Observations

**Lines 50-90:**

The discussion of SI3 physics is thorough but tends to conflate "detail" with "accuracy." Please avoid language such as "more accurate" when the evidence shown only supports "more detailed" physical representation of the ITD. Claims of improved accuracy should be supported by comparisons with independent observational datasets or independent models.

**Lines 90-120:**

Please clarify the rationale for the choice of observation products (OSI SAF datasets, Table 1). This could be as simple as noting that these datasets were considered state-of-the-art for their respective time periods. For example, while the text indicates that OSI-450-a, OSI-430-a, and OSI-438 were used sequentially, it does not discuss potential cross-calibration issues or bias transitions between these products. If such effects were evaluated and found negligible—or were deemed unquantifiable—please state this explicitly.

**Lines 110–115:**

Please improve the justification of the chosen NEMOVAR settings. Further explain why only a few parameters were modified, and reference relevant literature to support the selected background error covariance matrix configuration. What is the rationale for using a constant value of 0.2 for the observation operators? It is likely that a parabolic error distribution would be more appropriate for SIC, as it better represents lower uncertainty at the physical bounds (0 and 1) and higher uncertainty at intermediate concentrations.

3. Figures and Visualization

**Figures 1–8:**

Figures would benefit from clearer visual presentation. Specific points:

- **Figure 1:** Use darker ocean coloring and simplified snow overlays to better illustrate ITD structure and surface layering.
- **Figures 4–8:** Label all color bars and ensure aspect ratios are not distorted. Several figures appear stretched vertically.
- Figure 7: Avoid duplicate words (e.g., "sea sea")

- **Figure 8:** Clarify whether this map represents a single cycle or an average over multiple cycles.
- Please include the number of experiments or assimilation cycles used to produce each composite figure, either within the figure itself or in its caption. Without this information, it is difficult to assess the reproducibility and robustness of the presented results.

**4. Results and Interpretation**

**Lines 240–260 (Gamma and Peterson splitting tests):**

The text concludes that the proportional ("background") increment method performs best, due to worse performance from the gamma-based distribution in the Fram Strait (Fig. 4), which is important for the ECMWF NWP system. Was it ever considered to use the gamma-based distribution within the Central Arctic, which yields better SIC scores, and the "background" increment scheme near the ice edge? Also consider whether observational uncertainty or regional biases in the Level 3 product may have influenced these comparisons (especially in the marginal ice zone).

**Lines 260–290 ( $\alpha$ sensitivity tests):**

The selection of  $\alpha = 5$  as optimal appears partially ad hoc. Please include the rationale behind only testing  $\alpha \le 5$ . Mention whether  $\alpha > 5$  yields further improvement or instability.

**Lines 275–285 (Conclusions):**

The discussion asserts that "ice-induced temperature increments give the largest impact of all developments" based on your results, but the causal physical explanation is missing. Please elaborate/explore the physics in greater detail why this term may dominate the performance gain.

The conclusion is also much more of a discussion of future work as opposed to a wrap-up of the completed work. Perhaps a future work and/or discussion section focusing on this is desired.

**Minor Comments:**

**Introduction**

Line 18: "as a (thermo)dynamical isolating cover"  $\rightarrow$  "as a (thermo)dynamic insulating barrier" (more precise)

Line 34: "Other developments on sea-ice assimilation including attempt"  $\rightarrow$  "Other developments in sea-ice assimilation include attempts"

**Section 2**

Lines 59-60: "uses enthalpy rather than temperature, as in LIM2, as a prognostic variable" → Reorder: "uses enthalpy, as in LIM2, rather than temperature as a prognostic variable" (clearer)

**Section 3**

Line 77: Should equation (1) have a period after it?

**Section 4**

Line 124: "This is not a well defined problem"  $\rightarrow$  "This is not a well-defined problem" (hyphenate)

Line 126: "it is desirable that"  $\rightarrow$  "ideally," (more concise)

Line 145: "The net result of this is that"  $\rightarrow$  "Consequently,"

Line 159: "meltpond concentrations/volumes/lid volumes" → "melt pond concentrations, volumes, and lid volumes" (consistency and clarity)

Lines 160-161: "potentially because sea ice areal age is proportional to concentration, not volume" → Please rephrase this statement for clarity and explain how SI3 calculates sea ice areal age.

**Section 5**

Line 212: "Hourly data from ERAS5"  $\rightarrow$  "ERAS5" should be "ERA5"

Line 220: "we show results from deterministic (single member) experiments, using the ORAS6 EDA" → Clarify: how do the single members relate to the EDA? Please be more specific.

**Section 6**

Line 241: "The scores shown in Table 2 show that"  $\rightarrow$  "Table 2 shows that" (remove redundancy)

Line 252: "Global (err)" in table headers → Define "err" in caption – what does this error represent physically, and how does it relate to being a standard deviation?

**Section 7**

Line 271: "Sea ice concentration is an important dataset to be assimilated"  $\rightarrow$  "Sea ice concentration is an important variable to assimilate" (datasets vs. variables)

Line 296: "In the near future" → Specify timeframe

**General observations:**

- Inconsistent hyphenation: "sea-ice" vs "sea ice" standardize throughout
- "melt pond" vs "meltpond" be consistent
- Some sentences begin with "This" or "These" without clear antecedents consider being more specific

---

## Referee Comment (RC2)

This paper introduces new developments to the sea ice data assimilation in the ECMWF Ocean Reanalysis System. It particularly focuses on how to address the problem of implementing data assimilation increments into multicategory sea ice models. The experiments are well designed and thought out, and the results can be useful for sea ice data assimilation developments in the future. The article provides interesting and novel methods for improving the sea ice data assimilation method. However, the justification of some choices in the study as well as discussion of the results should be enhanced before publication. In addition, some of the figures need refining.

**Major Comments**

Why is OSISAF sea ice concentration chosen over other available observations? The choice of OSISAF data over other datasets needs to be justified in the text. It should also be considered how the observational uncertainty of the sea ice concentration observations effects the results particularly at the ice edge.

In figure 4 the authors only show the difference between the Gamma splitting and background splitting, but I think it would also be informative and useful for the discussion to show Peterson splitting against background splitting or Gamma splitting against Peterson splitting. This would be useful to see if there are further differences at the ice edge.

The ice induced temperature increment approach for the assimilation is interesting. I am wondering why the authors only chose smaller values of  $\alpha$  than the control for their experiments in Section 6.3, and none larger? From the current analysis it looks like an  $\alpha$  larger than five may lead to lower RMSE.

**Minor Comments**

Sometimes used sea-ice with hyphen and sometimes sea ice without hyphen, need to be consistent.

Line 24: NEMOVAR not defined

Line 28: Many others have applied Ensemble Kalman filter approaches which are relevant to this study:

Fritzner et al., 2019 assimilates SIC, SIT and snow depth in this approach

Williams et al., 2023 Assimilation of thickness distribution

Fiedler et al., 2023 Assimilation of Freeboard

As another method: Yang et al., 2014 assimilates sea ice using SEIK filter

Line 34: "Other developments on sea-ice assimilation including attempt to constrain" should be "include attempts to"

Line 50: remove "the" before SI3

Line 70: Are these the default bounds for the categories in the SI3 sea ice model or do the authors choose unique ones? Why are they chosen?

Line 106: "We have seen that" can be removed.

Line 126: Sea ice concentration and thickness are often treated as independent variables, but they can be correlated. For example during the melt season thinner ice will melt more quickly than thicker ice from the ocean and atmospheric forcings.

Line 201: should be "first appeared"

Line 216: "Is used" is repeated

Line 230: The sentence is unclear and quite long. It should be reworded or broken down for readability.

Line 274: The sentence "Ice induced temperature increments give the largest impacts of all developments." should be expanded upon/explained further to conclude clearly.

Lines 282-292: The work of Bocquet et al., 2024 which produced a longer time series of freeboard and ice thickness from 1994 for both Arctic (winter only) and Antarctic (year-round) should be mentioned also. Numerous other studies which have also assimilated ice thickness/freeboard alongside Mignac et al., 2022 could also be mentioned.

**Figures**

Generally, in many figures the colour bar is not labelled and units are not provided.

Figure 1: The authors used "melt pond" in the caption, but "meltpond" without space in the figure, need to be consistent.

Figure 5: It's difficult to see the differences with the current chosen colour scale.

Figure 7: "sea" is repeated in the caption.

**References**

Bocquet, M., Fleury, S., Rémy, F. and Piras, F., 2024. Arctic and Antarctic sea ice thickness and volume changes from observations between 1994 and 2023. *Journal of Geophysical Research: Oceans*, 129(11), p.e2023JC020848

Fiedler, E.K., Martin, M.J., Blockley, E., Mignac, D., Fournier, N., Ridout, A., Shepherd, A. and Tilling, R., 2022. Assimilation of sea ice thickness derived from CryoSat-2 along-track freeboard measurements into the Met Office's Forecast Ocean Assimilation Model (FOAM). *The Cryosphere*, 16(1), pp.61-85.

Fritzner, S., Graversen, R., Christensen, K.H., Rostosky, P. and Wang, K., 2019. Impact of assimilating sea ice concentration, sea ice thickness and snow depth in a coupled ocean—sea ice modelling system. *The Cryosphere*, *13*(2), pp.491-509.

Williams, N., Byrne, N., Feltham, D., Van Leeuwen, P.J., Bannister, R., Schroeder, D., Ridout, A. and Nerger, L., 2023. The effects of assimilating a sub-grid-scale sea ice thickness distribution in a new Arctic sea ice data assimilation system. *The Cryosphere*, *17*(6), pp.2509-2532.

Yang, Q., Losa, S.N., Losch, M., Tian-Kunze, X., Nerger, L., Liu, J., Kaleschke, L. and Zhang, Z., 2014. Assimilating SMOS sea ice thickness into a coupled ice-ocean model using a local SEIK filter. *Journal of Geophysical Research: Oceans*, *119*(10), pp.6680-6692.

---

## Referee Comment (RC3)

**Review of: Sea ice data assimilation in ORAS6**

by Phillip Browne, Eric de Boisseson, Sarah Keely, Charles Pelletier and Hao Zuo

December 3, 2025

**Manuscript Synopsis**

This paper is a description of the sea ice data assimilation process in ORAS6. While being an overall description of the sea ice data assimilation, the paper's main emphasis is to document the procedure of propagating the sea ice concentration increments into sea ice thickness category increments. The authors describe their method of distributing the total sea ice concentration increment to the thickness categories via equally distibuting it across existing categories very similarly to Smith et al. [2016]. They then demonstrate the improvement over an earlier strategy [Peterson et al., 2015] to only add or subtract the sea ice increment to the lowest available thickness category. Their demonstration of the improvement using their method is compelling, and on this basis alone I would recommend publication. However, I do have a comment concerning the explanation of perturbations employed to the sea ice observations (last paragraph of Section 3, ll. 100–104) to generate the ensemble analysis. While this does not have application for the results in the manuscript, which only use the control, unperturbed analysis, it may impact how the results will influence the wider ensemble, and aid in how they are interpreted.

**My recommendation is Minor Revisions**

**Major Comments**

1. The results from the manuscript are based on the non-perturbed control member of the ensemble ORAS6 system. Therefore the short existing description describing the perturbations to the sea ice concentration observations could be removed. As it stands, I found the ensemble perturbation description just informative enough to not understand what has been instituted. However, for selfish reasons – I really wish to understand this – I would prefer they firm up the description somewhat, even if it does not have direct bearing on the manuscript. The alternative might be to refer the reader to some other reference, either past or future (tech documents being acceptable).

   The tech document [Zuo et al., 2017] gives a detailed description of the random superobbing of OSI-SAF sea ice concentration used in ORAS5. I assume this corresponds to the one sentence statement (l.100) "The data is randomly thinned to boxes of 0.25/0.25 degrees."

   I have many questions regarding the further description detailing randomly sampling both the structural error determined by differences between OSTIA and ESA SST CCI v2, and the analysis error determined by variance in ERA5.

   - Are the structural and analysis errors actually independent. Doesn't ERA5 use OSTIA as the lower boundary condition?
   - Are the differences between OSTIA and ESA SST CCI v2 a good representation of the uncertainty in observed sea ice concentration. Do they not use the same OSI-SAF sea ice product, or at least similar OSI-SAF products? I researched this, OSTIA uses the near realtime OSI-401-b and ESA SST CCI v2 presumably (no clear link to which product used) uses one of the climate reanalysis products of OSI-450-a1 or OSI-430-a. These will differ, but only by the choice of the long term anchor mechanism use to produce a self-consistent climate product, but not in the actual retrieval method.

- How do you build a database from each of these? I envision this could be as small a database as two numbers – a difference, and a variance. However, I suspect it must be a much fuller database that somehow incorporates temporal and spatial covariances into it. A more detailed explanation, or reference to a more detailed explanation of what is done here would be beneficial.

Why this might be important: There is a large uncertainty between the sea ice analysis products [e.g. Peterson et al., 2022, Niraula, 2023]. This large uncertainty, and in particular the fact that the sea ice edge from the OSI-SAF products appears to be too diffuse Renfrew et al. [2021], could have an impact on your results.

I want to be clear: I completely agree that it is best to test the methodologies for distributing the sea ice into thickness categories in the control, unperturbed, deterministic simulation, and then validating the sea ice concentration fit against the assimilated observations. (I.e. I am not suggesting any changes to your methodology.) However, I believe results such as the degraded (Atlantic) sea ice edge thickness (Figure 4) when employing the gamma thickness distribution are likely related to uncertainties in the ice concentration observations – and this should probably be acknowledged. Similarly, the need to suppress temperature increments at the expense of sea ice increments (Section 6.3) is again likely a result of uncertainties in both the sea ice observations and SST analysis close to the sea ice edge as covered in my next point.

2. Section 6.3: I have a small issue with referring to the suppression of temperature increments as a balance relationship. For a balance constraint, typically one would have a balanced temperature increment dependent on the sea ice, but then an additional unbalanced term and unbalanced increments to temperature that could still improve the fit to temperature observations. But more philosophically, here you are literally placing your finger on the *balance scale* by tipping the resulting analysis fit towards the sea ice observations and away from the temperature observations. If anything, it would be more akin to introducing bogus (zero innovation) observations in the presense of sea ice. All these methods, along with still others are employed to improve retention of increments in the analysis, so I have no issue with the method, just perhaps the terminology. However, my understanding is not complete, and maybe they are all interconnected.

When first I looked at the results it is clear that the fit to sea ice does improve with increasing alpha – but it was not clear that your choice of alpha was actually maximizing that fit (i.e. would it be a good idea to test a higher value of alpha). With better understanding, I now believe there would be no maximization (although perhaps saturation); continuing to increase alpha should continue to increase the fit to the sea ice observations, although presumably in detriment to the fit to temperature observations and presumably with lowering levels of improvement as alpha is further increased. What is shown (Tables 4 &5; Figures 6,7 &8) is only half the story as presumably this is all at the cost of sea surface and sub-surface temperature statistics (but only in the vicinity of the sea ice edge).

This then connects with item 1: How does this connect with uncertainties in the sea ice concentration and temperature observations? Are you improving a fit to observation (in reality, that "observation" is actually an analysis) to a point the fit is well below the significant uncertainty in the observation? Figure 7, 8 & 9 of Renfrew et al. [2021] would seem to suggest this might be so. Is there a need to re-address this issue with post 2025-07-06 results and the introduction of the OSI-438 (AMSR2) product (Table 1)?

**Minor Comments**

1. I think it should be pointed out other methods of distributing sea ice concentration amongst the thickness categories do exist. For example, Smith et al. [2016] propose the rescaled forecast tendencies (RFT) method that is largely identical to your method. For instance their Eqn. 1 is identical to your equation 3. Please cite.

2. Joint Minimization (l. 114): It has been my understanding that joint minimimization of the univariate sea ice and multivariate ocean has always been a problem that leads to degraded fits, so I am happy that

you have managed to achieve this. However, will this not be a function of the observing system. Could improved numbers of near ice temperature observations change this dynamic? I suppose the hope is to go toward a more balanced, joint multi-variate ocean sea-ice assimilation before any substantial changes occur in the observing network.

3. Figure 4 (Section 6.1). As stated above in Major Item #1, I believe the degradation in sea ice thickness when using the gamma distribution is ultimately due to the uncertainties in the sea ice concentration and location of the ice edge. Although we are given insufficient information to confirm this, I suspect the gamma distribution has a bias towards thicker ice, at least at the ice edge (what is the hemisphere change in bias?) – which then shows up as increased RMSE there as the thickness observations would be sensing zero thickness ice. (I.e. Your decision – I believe correct decision – to not utilize the slightly better in terms of ice concentration, gamma distribution, is likely due to offsetting biases.) This might be a result that could change in the perturbed solution. It also again would likely change post 2025-07-06 with the introduction of OSI-438.

4. Open Water (Section 6.2) I could not help but think your open water fit results were one-sided. The results continue to get better as you increase the thickness ice increments in open water. While I would agree adding ice to open water into the 2nd category would seem weird – and would disconnect your data assimilation new ice thickness to your thermodynamic new ice thickness. (Actually that wording is a little ambigouous in Section 4.3.2 – the new ice thickness in the DA is or is not identical to the thermodynamic new (frazil) ice – or just the other properties?) At any rate, the same arguments that the data assimilation process is correcting for errors in dynamical movement of ice as much as errors in thermodynamics as justification for adding ice across all categories could presumably be used to justify adding higher category (relocated ice) for positive increments in open ice. Sorry: I seem to recall a statement is made along these lines, but I could not track it down in the manuscript. It might have been interesting to test adding 2nd category thickness ice to see where the fit to observations begins to decrease.

5. Table 4&5/Figure 8: If I have not missed something, $\alpha$ (Eqn. 5) has units of °C or K (it is a change in temperature – so your choice). The values of $\alpha$ in Tables 4&5 and Figure 8 – and in the text if they occur there – should be accompanied with that unit.

P.S. I will forego my anonymity as penance for my tardiness in achieving this review.

**References**

B. Niraula. *Ice Edge Verification – Measuring the skill in our forecasts and disagreement in our observations*. PhD thesis, Universität Bremen, 2023. URL `https://doi.org/10.26092/elib/2298`.

K. Andrew Peterson, A. Arribas, H.T. Hewitt, A.B. Keen, D.J. Lea, and A.J. McLaren. Assessing the forecast skill of Arctic sea ice extent in the GloSea4 seasonal prediction system. *Climate Dynamics*, 44 (1-2):147–162, 2015. ISSN 0930-7575. doi: 10.1007/s00382-014-2190-9. URL `http://dx.doi.org/10.1007/s00382-014-2190-9`.

K. Andrew Peterson, Gregory C. Smith, Jean-François Lemieux, François Roy, Mark Buehner, Alain Caya, Pieter L. Houtekamer, Hai Lin, Ryan Muncaster, Xingxiu Deng, Frédéric Dupont, Normand Gagnon, Yukie Hata, Yosvany Martinez, Juan Sebastian Fontecilla, and Dorina Surcel-Colan. Understanding sources of northern hemisphere uncertainty and forecast error in a medium-range coupled ensemble sea-ice prediction system. *Quarterly Journal of the Royal Meteorological Society*, 148(747):2877–2902, 2022. doi: https://doi.org/10.1002/qj.4340. URL `https://rmets.onlinelibrary.wiley.com/doi/abs/10.1002/qj.4340`.

I. A. Renfrew, C. Barrell, A. D. Elvidge, J. K. Brooke, C. Duscha, J. C. King, J. Kristiansen, T. Lachlan Cope, G. W. K. Moore, R. S. Pickart, J. Reuder, I. Sandu, D. Sergeev, A. Terpstra, K. Våge, and A. Weiss. An evaluation of surface meteorology and fluxes over the iceland and greenland seas in ERA5 reanalysis: The impact of sea ice distribution. *Quarterly Journal of the Royal Meteorological Society*, 147(734):691–712, 2021. doi: https://doi.org/10.1002/qj.3941. URL `https://rmets.onlinelibrary.wiley.com/doi/abs/10.1002/qj.3941`.

Gregory C. Smith, François Roy, Mateusz Reszka, Dorina Surcel Colan, Zhongjie He, Daniel Deacu, Jean-Marc Belanger, Sergey Skachko, Yimin Liu, Frédéric Dupont, Jean-François Lemieux, Christiane Beaudoin, Benoit Tranchant, Marie Drévillon, Gilles Garric, Charles-Emmanuel Testut, Jean-Michel Lellouche, Pierre Pellerin, Harold Ritchie, Youyu Lu, Fraser Davidson, Mark Buehner, Alain Caya, and Manon Lajoie. Sea ice forecast verification in the Canadian Global Ice Ocean Prediction System. *Quarterly Journal of the Royal Meteorological Society*, 142(695):659–671, 2016. doi: 10.1002/qj.2555. URL `https://rmets.onlinelibrary.wiley.com/doi/abs/10.1002/qj.2555`.

Hao Zuo, Magdalena Alonso-Balmaseda, Eric de Boisseson, S Hirahara, Marcin Chrust, and Patricia de Rosnay. A generic ensemble generation scheme for data assimilation and ocean analysis. Technical report, ECMWF, 2017. URL `https://www.ecmwf.int/node/17831`.

---

## Author Comment (AC1)

**Response to reviewer 1**

December 1, 2025

Thank you to the reviewer for taking the time to provide excellent constructive comments which should help improve the manuscript.

Major Comments

1. Conceptual and Methodological Clarity

- Lines 15–40+: Please define all acronyms (ECMWF, ORAS6, LIM2, SI3, NEMOVAR, etc.) upon first mention. The introduction assumes strong familiarity with ECMWF's system architecture, which may limit accessibility for broader readership.

  Indeed there are some missing definitions which we will gladly expand.

- Lines 120–135:

  The statement in Line 126 that "sea ice concentration is orthogonal to sea ice thickness" requires careful revision. These quantities are not strictly orthogonal in the linear algebraic sense; rather, they are inherently linked through the ice volume relationship $SIV_n = SIC_n \times SIT_n$. For instance, during the melt season, increased atmospheric and oceanic temperatures cause thinner ice to melt more rapidly, leading to a decrease in SIC. Meanwhile, thicker ice (i.e., higher SIT) tends to persist longer, resulting in spatial patterns where SIC and SIT covary—contradicting the claim of orthogonality. During the freeze-up period, SIC and SIT may indeed covary less, but as illustrated above, this relationship is not universally negligible. Assuming zero covariance between the two may therefore misrepresent coupled ice processes and potentially lead to unphysical SIT (and consequently SIC) states. While SIC and SIT are often treated as distinct state variables, it is inaccurate to describe them as orthogonal. This assumption should be revisited, and the implications for the chosen increment distribution scheme should be discussed. If SIC is updated independently across categories primarily for practical reasons, that rationale should be made explicit.

  We understand the reviewer's strong view here, and agree with all their scientific points. However we think this is partially a miscommunication on our part, as we have used the term "orthogonal" in the geometric sense to mean perpendicular. We absolutely agree that the quantities are related and correlated, especially with a multicategory model, but they remain representative of spatial extent (SIC) and vertical extent (SIT). This we attempt convey in Figure 1.

  We would propose to change the wording such that our statement "Noting that sea ice concentration is orthogonal to sea ice thickness" would become "Noting that sea ice concentration represents the spatial extent and sea ice thickness represents the vertical extent"

- Following up on this point (and similar to the one above), since sea ice volume is an extensive variable and is updated as such in Equation 4 (Line 158), have you considered how restricting

changes in the ITD impacts your updates? For example, imagine an update in the Central Arctic, where ice is relatively thick ($< 1$ m). Suppose a ridging event occurs that leaves open water, and the ice then refreezes according to observations and the prevailing cold atmospheric state, warranting a positive SIC increment. This newly formed ice is likely thin to start, according to well-established theory, but under your current update scheme, it would be constrained to adhere to the original thickness distribution—potentially overcompensating and producing too much ice volume, leading to an unphysical amount of ice. Have you considered this or similar scenarios? the ice volume increment could be scaled more strongly toward thinner ice categories in both positive and negative increments to better represent the persistence of thicker ice, as widely documented in the literature.

This is a very clear example of a limitation of our system. In this case we rely completely on the model dynamics to simulate such events, and if they are missed then the resulting analysis will become biased. What we need in order to constrain our system are observations of the sea ice thickness. We lack the capacity to assimilate available sea ice thickness observations and are working actively on their inclusion in a future system. (See for example, the WMO WWRP PCAPS task team on sea ice thickness data assimilation https://www.wwrp-pcaps.net/what-we-do#task-teams, ESA funded DANTEX project https://www.ecmwf.int/en/elibrary/81674-earth-system-assimilation-cryosphere-status-and-way-forward-td-01). We assimilate SIC observations in our multicategory model so that the category-average thickness is left unaltered, partly because we did not want SIC observations to bear an impact on effective thickness. Our hope is that these future SIT related observations will then be able to constrain that other dimension to the representation of the sea ice. We also note that the scheme of Peterson et al. (2015) goes some way to implement the methodology you suggest, which we tested (line 230-245) and showed to perform less well than Gamma or background splitting.

- Lines 165–175: The approach of assigning a fixed thickness of 45 cm to newly formed ice is insufficiently justified. The sensitivity tests in Section 6.2, which explore the additional fixed thicknesses of 22.5 cm and 10 cm, are valuable but not comprehensive. A more physically consistent approach would be to assign thickness proportionally to the magnitude of the SIC increment (e.g., thinner ice—approximately 10 cm—for small increments and thicker ice—up to 45 cm—for larger increments). Please also provide the specific reference supporting the 0.5 m threshold used in ORAS5.

We cannot go any thicker than 0.45m as adding ice into categories that are not the thinnest poses technical challenges which we were unable to overcome. What we have shown with the sensitivity tests is that we need to add ice with as much enthalpy as possible to stop the model from melting out the new ice. Thus any weighting by SIC increment would act to give less enthalpy to the new ice, and thus the preexisting biases in the ocean state and atmospheric forcing would more easily remove the ice that the observations are informing should be there. The 0.5m minimum SIT was not documented in the ORAS5 paper. This is the same value used by LIM2 model when forming new sea-ice in the open water. We understand that the same 0.5m of SIT was used in the UK Met Office system when new ice is added.

We are happy to add a reference to Fichefet and Maqueda [1997] where they also use 0.5m as the thickness of new ice.

T. Fichefet and M. M. Maqueda. Sensitivity of a global sea ice model to the treatment of ice thermodynamics and dynamics. *Journal of Geophysical Research: Oceans*, 102(C6):12609–12646, 1997

- Lines 180–195: The implementation of an "ice-induced temperature increment" is a creative solution to maintain thermodynamic balance, but its physical basis and vertical extent (to 19.5 m) are questionable. Please provide justification for the chosen $\alpha$ values and the depth profile f(z), as decreasing ocean temperature at such depth could unintentionally alter stratification or vertical mixing. It is unclear whether this adjustment differentiates between increments arising from advection versus thermodynamics—this distinction should be addressed. For example, in the case of advection, atmospheric forcing from cyclones may move ice equatorward, which most likely does not alter the near-surface ocean profile in the same way that the thermodynamic growth of ice would.

  We agree that the physical basis for this mechanism is not carefully considered . Indeed, it should not be needed if the ocean temperatures were perfect and conducive to maintaining ice of the required concentration at the surface. Ideally further developments could be made in this area, such as ensuring the energy content of the temperature increment is somehow related to the latent heat of fusion. As explained, ideally such an approach should be achieved within the balance operator of data assimilation scheme, which is not yet implemented in our system. Various options of alpha have been tested (see Chapter 6.3). However, we do acknowledge that the final choice of this vertical profile f(z) and value of alpha could be refined further, based on physical processing and exchange of energy between liquid water and sea-ice.

  There is no case where we differentiate between increments arising due to model deficiencies in either advection or thermodynamics. Untangling which scenario we are in from observations of SIC alone seems like a challenging problem to automate for every cycle of a reanalysis.

- Lines 220–270: The experimental design is clearly described; however, several key methodological details are missing and appear to rely on the reader's prior familiarity with ORAS6. Please specify the number of ensemble members at the first mention (e.g., near Line 100, rather than only at Line 216). Further clarification is also needed regarding the "deterministic (single-member) experiments" described at Line 220—specifically, how they were implemented, why this approach was chosen, and whether any statistical significance testing was performed to account for potential control-member dependence (e.g., rerunning with five different control members to assess robustness). Additionally, the presentation of normalized standard deviations in Tables 2– 5 would be clearer if accompanied by absolute RMSE values and the sample size used for each experiment (e.g., n = …).

  We are happy to provide these clarifications. We would add the deterministic method is chosen for computational cost reasons given the high expense of running a global ocean/sea ice reanalysis for the 5 year period. We are unsure how to describe "how they were implemented" in a way that adds any meaningful insight. No significance testing was performed to account for potential control-member dependence.

  We have approximately 250,000 observations per cycle, and the experiments run for 360 cycles over the 5 year period, leading to a total number of observations around n = 90,000,000. We can add that the control standard deviations are globally: 0.098 $m^2/m^2$ with a sample size of $9.15 \times 10^7$, northern hemisphere 0.130 $m^2/m^2$ with a sample size of $3.82 \times 10^7$, southern hemisphere 0.095 $m^2/m^2$ with a sample size of $5.32 \times 10^7$,

2. Physical Justification and Link to Observations

- Lines 50–90:

  The discussion of SI3 physics is thorough but tends to conflate "detail" with "accuracy." Please avoid language such as "more accurate" when the evidence shown only supports "more

detailed" physical representation of the ITD. Claims of improved accuracy should be supported by comparisons with independent observational datasets or independent models.

We are happy to remove statements about accuracy here, we do not want this paper to be a reference for the model performance comparing SI3 with LIM so it is right that we reword things here.

- Lines 90–120: Please clarify the rationale for the choice of observation products (OSI SAF datasets, Table 1). This could be as simple as noting that these datasets were considered state-of-the-art for their respective time periods. For example, while the text indicates that OSI-450-a, OSI-430-a, and OSI-438 were used sequentially, it does not discuss potential cross-calibration issues or bias transitions between these products. If such effects were evaluated and found negligible—or were deemed unquantifiable—please state this explicitly.

As in response to RC2:

We move to L3 because the assimilation system can cope with missing data, something the old system could not do and therefore required L4 data (gap filled). OSISAF provides a climate data record from 1978 onwards which seamlessly transitions into a near real time product. This near real time product is suitable for the operational schedule for NWP production. Moreover the data is available with operational levels of support. We have decided to move the contexts about OSI-SAF data transition to fast track data and OSI-438 product to the ORAS6 reference (currently in preparation), including corresponding discussion on the cross-calibration and continuation of sea-ice states during these transition.

- Lines 110–115: Please improve the justification of the chosen NEMOVAR settings. Further explain why only a few parameters were modified, and reference relevant literature to support the selected background error covariance matrix configuration. What is the rationale for using a constant value of 0.2 for the observation operators? It is likely that a parabolic error distribution would be more appropriate for SIC, as it better represents lower uncertainty at the physical bounds (0 and 1) and higher uncertainty at intermediate concentrations. As noted we have kept all the settings listed in this section the same as implemented in ORAS5. We agree with the reviewer that these settings, particularly, the specification of observation errors for sea-ice concentration, could be further refined, e.g. using the parabolic distribution as suggested. We will add a sentence to clarify this point. All of these could be optimised but have not been.

3. Figures and Visualization Figures 1–8: Figures would benefit from clearer visual presentation. Specific points:

- Figure 1: Use darker ocean coloring and simplified snow overlays to better illustrate ITD structure and surface layering.

Figures 4–8: Label all color bars and ensure aspect ratios are not distorted. Several figures appear stretched vertically.

Figure 7: Avoid duplicate words (e.g., "sea sea")

Figure 8: Clarify whether this map represents a single cycle or an average over multiple cycles.

Please include the number of experiments or assimilation cycles used to produce each composite figure, either within the figure itself or in its caption. Without this information, it is difficult to assess the reproducibility and robustness of the presented results.

We are happy to correct and/or clarify these. Unless otherwise stated (figures 6 & 7), the plots are from the entire 5 year experimental period. We can add that Figure 2 is an instantaneous look at the model field.

4. Results and Interpretation

- Lines 240–260 (Gamma and Peterson splitting tests): The text concludes that the proportional ("background") increment method performs best, due to worse performance from the gamma-based distribution in the Fram Strait (Fig. 4), which is important for the ECMWF NWP system. Was it ever considered to use the gamma-based distribution within the Central Arctic, which yields better SIC scores, and the "background" increment scheme near the ice edge? Also consider whether observational uncertainty or regional biases in the Level 3 product may have influenced these comparisons (especially in the marginal ice zone).

  No this was not considered. It is certainly an interesting suggestion which we can consider for future work.

- Lines 260–290 ($\alpha$ sensitivity tests): The selection of $\alpha = 5$ as optimal appears partially ad hoc. Please include the rationale behind only testing $\alpha \leq 5$. Mention whether $\alpha > 5$ yields further improvement or instability.

  As in response to RC2:

  This is a very fair question. We are conservative in the approach and do not want the ice induced temperature increments to be larger as they could be compensating for errors in the forcing and not in the ocean state. Following equation (5), if $\delta a$ is, say, 0.5 (a 50% change in ice concentration - large but not unexpected near the ice edge), then with $\alpha = 5$ this gives a 2.5K temperature increment to the ocean. These are regions where there are limited in situ temperature measurements at depth, and so we fear making even larger temperature increments could have negative consequences on the 3D ocean state. We do hope to improve on the simplicity of this scheme in the future, and very much recognise the need for and welcome suggestions for future developments to this approach.

- Lines 275–285 (Conclusions): The discussion asserts that "ice-induced temperature increments give the largest impact of all developments" based on your results, but the causal physical explanation is missing. Please elaborate/explore the physics in greater detail why this term may dominate the performance gain. The conclusion is also much more of a discussion of future work as opposed to a wrap-up of the completed work. Perhaps a future work and/or discussion section focusing on this is desired.

  We can certainly expand on this. We will speculate that, due to the lack of observational temperature constraints below the sea ice, model biases exist which prevent the ice model from either forming or sustaining sea ice of the concentration detected by the observations. Hence the ice-induced temperature increments are going some way to counteract this model bias.

The following minor comments are all well noted and we are happy to address in a revised version of the manuscript.

Minor Comments:

- Line 18: "as a (thermo)dynamical isolating cover" → "as a (thermo)dynamic insulating barrier"

- Line 34: "Other developments on sea-ice assimilation including attempt" → "Other developments in sea-ice assimilation include attempts"

  Section 2

- Lines 59-60: "uses enthalpy rather than temperature, as in LIM2, as a prognostic variable" → Reorder: "uses enthalpy, as in LIM2, rather than temperature as a prognostic variable" (clearer)

Section 3

- Line 77: Should equation (1) have a period after it?

Section 4

- Line 124: "This is not a well defined problem" → "This is not a well-defined problem" (hyphenate)

- Line 126: "it is desirable that" → "ideally," (more concise)

- Line 145: "The net result of this is that" → "Consequently,"

- Line 159: "meltpond concentrations/volumes/lid volumes" → "melt pond concentrations, volumes, and lid volumes" (consistency and clarity)

- Lines 160-161: "potentially because sea ice areal age is proportional to concentration, not volume" → Please rephrase this statement for clarity and explain how SI3 calculates sea ice areal age.

Section 5

- Line 212: "Hourly data from ERAS5" → "ERAS5" should be "ERA5"

- Line 220: "we show results from deterministic (single member) experiments, using the ORAS6 EDA" → Clarify: how do the single members relate to the EDA? Please be more specific.

Section 6 redundancy)

- Line 241: "The scores shown in Table 2 show that" → "Table 2 shows that" (remove

- Line 252: "Global (err)" in table headers → Define "err" in caption – what does this error represent physically, and how does it relate to being a standard deviation?

Section 7

- Line 271: "Sea ice concentration is an important dataset to be assimilated" → "Sea ice concentration is an important variable to assimilate" (datasets vs. variables)

- Line 296: "In the near future" → Specify timeframe

General observations:

- Inconsistent hyphenation: "sea-ice" vs "sea ice" - standardize throughout

- "melt pond" vs "meltpond" - be consistent

- Some sentences begin with "This" or "These" without clear antecedents - consider being more specific

---

## Author Comment (AC2)

**Response to reviewer 2**

December 1, 2025

Thank you to the reviewer for taking the time to provide excellent constructive comments which should help improve the manuscript.

- "Why is OSISAF sea ice concentration chosen over other available observations? The choice of OSISAF data over other datasets needs to be justified in the text."

  As in response to RC1:

  On the choice of OSISAF observations: there are a number of reasons why we assimilate the OSISAF L3 data. We move to L3 because the assimilation system can cope with missing data, something the old system could not do and therefore required L4 data (gap filled). OSISAF provides a climate data record from 1978 onwards which seamlessly transitions into a near real time product. This near real time product is suitable for the operational schedule for NWP production. Moreover the data is available with operational levels of support. More information on the choice of observational data will be part of the overarching ORAS6 paper, currently in preparation.

- "It should also be considered how the observational uncertainty of the sea ice concentration observations effects the results particularly at the ice edge."

  The uncertainty associated with the product is not something that we currently account for. This will be addressed in future updates, but was not part of the production system due to the need to have finished production with tight time constraints. Observations are perturbed, as mentioned at the end of Section 3.1. However we realise this is a deficiency of the system and results in underestimation of the uncertainty in the ice edge in the ORAS6 ensemble.

- "In figure 4 the authors only show the difference between the Gamma splitting and background splitting, but I think it would also be informative and useful for the discussion to show Peterson splitting against background splitting or Gamma splitting against Peterson splitting. This would be useful to see if there are further differences at the ice edge."

  Thank you for asking for this plot - this has highlighted that we included the incorrect plot in our manuscript. Below we show the erroneous Figure 4, noting that it was actually showing the difference in performance in the Peterson splitting and the background splitting.

  Figure 4 CORRECTED, shows now gamma splitting in comparison to background splitting. We see that the same pattern exists - a large degradation in the Atlantic sector. We will happily replace the figure with corrected version, and show both in a new version of the manuscript.

  Moreover we will include Figure NEW to show verification in sea ice concentration against the ESA CCIv3 SIC product of Embury et al. [2024] . We think this will significantly help the reader understand the choices made regarding the splitting methodology.

[Figure]

Figure 4: Errors in analysis of sea ice thickness comparing to *CS2SMOS merged CryoSat-2 and SMOS ice thickness*. Shown is the difference in RMSE using the  Peterson splitting compared to background splitting. In red are regions of worse sea ice thickness, which is notable around the ice edge in the Atlantic.

O. Embury, C. J. Merchant, S. A. Good, N. A. Rayner, J. L. Høyer, C. Atkinson, T. Block, E. Alerskans, K. J. Pearson, M. Worsfold, et al. Satellite-based time-series of sea-surface temperature since 1980 for climate applications. *Scientific Data*, 11(1):326, 2024

[Figure]

Figure 4 CORRECTED: Errors in analysis of sea ice thickness comparing to *CS2SMOS merged CryoSat-2 and SMOS ice thickness*. Shown is the difference in RMSE using (a) the Peterson splitting, and (b) the Gamma splitting, compared to background splitting. In red are regions of worse sea ice thickness, which is notable around the ice edge in the Atlantic.

[Figure]

Figure NEW: Errors in analysis of sea ice concentration comparing to *ESA CCIv3*. Shown is the difference in RMSE using (a) the Peterson splitting, and (b) the Gamma splitting, compared to background splitting. In red are regions of worse sea ice concentration, which is notable across large areas of the Arctic domain when using Peterson splitting.

- "The ice induced temperature increment approach for the assimilation is interesting. I am wondering why the authors only chose smaller values of $\alpha$ than the control for their experiments in Section 6.3, and none larger? From the current analysis it looks like an $\alpha$ larger than five may lead to lower RMSE."

  As in response to RC1:

  This is a very fair question. We are conservative in the approach and do not want the ice induced temperature increments to be larger as they could be compensating for errors in the forcing and not in the ocean state. Following equation (5), if $\delta a$ is, say, 0.5 (a 50% change in ice concentration - large but not unexpected near the ice edge), then with $\alpha = 5$ this gives a 2.5K temperature increment to the ocean. These are regions where there are limited in situ temperature measurements at depth, and so we fear making even larger temperature increments could have negative consequences on the 3D ocean state. We do hope to improve on the simplicity of this scheme in the future, and very much recognise the need for and welcome suggestions for future developments to this approach.

- "Line 70: Are these the default bounds for the categories in the SI3 sea ice model or do the authors choose unique ones? Why are they chosen?"

  These are the bounds provided by the configuration with 5 categories which we take as our default, yes. It is entirely configurable - we know for instance at Mercator Ocean International they run with 11 ice categories with different bounds [Chenal et al., 2024]. There is some evidence that 5 categories are sufficent for different use cases, for example "It is also found that the current default discretization of the NEMO3.6-LIM3 model is sufficient for large-scale present-day climate applications.", [Massonnet et al., 2019].

  "With five to seven categories the errors due to finite resolution of the thickness distribution are much smaller than the errors due to other sources. These results suggest that seven categories are sufficient for climate modeling. Five categories are adequate ifwe accept errors of 0.1 W m -2 in the surface fluxes, 10 kN m - in the ice strength, and 2 cm in the thickness range." [Lipscomb, 2001].

  A. Chenal, G. Garric, C.-E. Testut, M. Hamon, G. Ruggiero, F. Garnier, and P.-Y. Le Traon. Assimilation of radar freeboard and snow altimetry observations in the arctic and antarctic with a coupled ocean/sea ice modelling system. *EGUsphere*, 2024:1–39, 2024. doi: 10.5194/egusphere-2024-3633. URL https://egusphere.copernicus.org/preprints/2024/egusphere-2024-3633/

  F. Massonnet, A. Barthélemy, K. Worou, T. Fichefet, M. Vancoppenolle, C. Rousset, and E. Moreno-Chamarro. On the discretization of the ice thickness distribution in the NEMO3.6-LIM3 global ocean–sea ice model. *Geoscientific Model Development*, 12(8):3745–3758, 2019. doi: 10.5194/gmd-12-3745-2019. URL https://gmd.copernicus.org/articles/12/3745/2019/

  W. H. Lipscomb. Remapping the thickness distribution in sea ice models. *Journal of Geophysical Research: Oceans*, 106(C7):13989–14000, 2001. doi: https://doi.org/10.1029/2000JC000518. URL https://agupubs.onlinelibrary.wiley.com/doi/abs/10.1029/2000JC000518

- "Line 126: Sea ice concentration and thickness are often treated as independent variables, but they can be correlated. For example during the melt season thinner ice will melt more quickly than thicker ice from the ocean and atmospheric forcings."

  As in response to RC1:

We understand the reviewer's strong view here, and agree with all their scientific points. However we think this is partially a miscommunication on our part, as we have used the term "orthogonal" in the geometric sense to mean perpendicular. We absolutely agree that the quantities are related and correlated, especially with a multicategory model, but they remain representative of spatial extent (SIC) and vertical extent (SIT). This we attempt convey in Figure 1.

We would propose to change the wording such that our statement "Noting that sea ice concentration is orthogonal to sea ice thickness" would become "Noting that sea ice concentration represents the spatial extent and sea ice thickness represents the vertical extent"

We recognise the remainder of the minor comments (below). We will happily correct/act on them in the manuscript.

- Sometimes used sea-ice with hyphen and sometimes sea ice without hyphen, need to be consistent.

- Line 24: NEMOVAR not defined

- Line 28: Many others have applied Ensemble Kalman filter approaches which are relevant to this study:

- Fritzner et al., 2019 assimilates SIC, SIT and snow depth in this approach

- Williams et al., 2023 Assimilation of thickness distribution

- Fiedler et al., 2023 Assimilation of Freeboard

- As another method: Yang et al., 2014 assimilates sea ice using SEIK filter

- Line 34: "Other developments on sea-ice assimilation including attempt to constrain" should be "include attempts to"

- Line 50: remove "the" before SI3

- Line 106: "We have seen that" can be removed.

- Line 126: Sea ice concentration and thickness are often treated as independent variables, but they can be correlated. For example during the melt season thinner ice will melt more quickly than thicker ice from the ocean and atmospheric forcings.

- Line 201: should be "first appeared"

- Line 216: "Is used" is repeated

- Line 230: The sentence is unclear and quite long. It should be reworded or broken down for readability.

- Line 274: The sentence "Ice induced temperature increments give the largest impacts of all developments." should be expanded upon/explained further to conclude clearly.

- Lines 282-292: The work of Bocquet et al., 2024 which produced a longer time series of freeboard and ice thickness from 1994 for both Arctic (winter only) and Antarctic (year-round) should be mentioned also. Numerous other studies which have also assimilated ice thickness/freeboard alongside Mignac et al., 2022 could also be mentioned.

- Figures Generally, in many figures the colour bar is not labelled and units are not provided.

- Figure 1: The authors used "melt pond" in the caption, but "meltpond" without space in the figure, need to be consistent.

- Figure 5: It's difficult to see the differences with the current chosen colour scale.

- Figure 7: "sea" is repeated in the caption.

---

## Author Comment (AC3)

**Response to reviewer 3**

**December 12, 2025**

Thank you to the reviewer for taking the time to provide excellent constructive comments which should help improve the manuscript.

Major Comments

- The results from the manuscript are based on the non-perturbed control member of the ensemble ORAS6 system. Therefore the short existing description describing the perturbations to the sea ice concentration observations could be removed. As it stands, I found the ensemble perturbation description just informative enough to not understand what has been instituted. However, for selfish reasons – I really wish to understand this – I would prefer they firm up the description somewhat, even if it does not have direct bearing on the manuscript. The alternative might be to refer the reader to some other reference, either past or future (tech documents being acceptable). The tech document [Zuo et al., 2017] gives a detailed description of the random superobbing of OSISAF sea ice concentration used in ORAS5. I assume this corresponds to the one sentence statement (l.100) "The data is randomly thinned to boxes of 0.25/0.25 degrees." I have many questions regarding the further description detailing randomly sampling both the structural error determined by differences between OSTIA and ESA SST CCI v2, and the analysis error determined by variance in ERA5.

    - Are the structural and analysis errors actually independent. Doesn't ERA5 use OSTIA as the lower boundary condition?
      ERA5 uses HadISST2 until 2008 and then OSTIA NRT

    - Are the differences between OSTIA and ESA SST CCI v2 a good representation of the uncertainty in observed sea ice concentration. Do they not use the same OSI-SAF sea ice product, or at least similar OSI-SAF products? I researched this, OSTIA uses the near realtime OSI-401-b and ESA SST CCI v2 presumably (no clear link to which product used) uses one of the climate reanalysis products of OSI-450-a1 or OSI-430-a. These will differ, but only by the choice of the long term anchor mechanism use to produce a self-consistent climate product, but not in the actual retrieval method.
      This is true. See below for more on this.

    - How do you build a database from each of these? I envision this could be as small a database as two numbers – a difference, and a variance. However, I suspect it must be a much fuller database that somehow incorporates temporal and spatial covariances into it. A more detailed explanation, or reference to a more detailed explanation of what is done here would be beneficial.
      See below.

Why this might be important: There is a large uncertainty between the sea ice analysis products [e.g. Peterson et al., 2022, Niraula, 2023]. This large uncertainty, and in particular

the fact that the sea ice edge from the OSI-SAF products appears to be too diffuse Renfrew et al. [2021], could have an impact on your results.

We understand your frustrations here and think some more context would be useful. The full ensemble perturbation strategy will be part of the up-coming ORAS6 reference paper. The short description we include here to keep such sea ice concentration details together. So we will ensure that a new version includes clear direction to either the new ORAS6 reference if it is submitted in time, or to the ORAS5 paper as you noted.

More importantly we already know that our perturbation strategy is insufficient to achieve realistic spread. In no large part this is because, as you have pointed out, OSTIA and ESA SST CCI v2 are based on similar underlying OSISAF product. This feature of the system was identified after production was completed and so we had no time to correct it with any other methodologies.

We wanted to keep this paper focused on the assimilation methodology which we would apply regardless of observation perturbation method. So we should acknowledge that the uncertainty (and how to account for it) in SIC observations could affect our assessments, and they be viewed with this in mind.

And as a further note, the lack of spread does not directly influence the data assimilation methodology we are using, as the spread is not used to estimate any flow dependent background errors for the sea ice. This is different to the 3D ocean, where background errors are derived from the ensemble spread. Work is planned for future systems to improve ensemble reliability in this area, both with new observation perturbation strategies, but also with stochastic physics developments in the model itself.

The following is a short, but more detailed description of the construction of the perturbation database. We prefer to leave such details out of this paper, and leave them for the overarching ORAS6 reference paper, but does no harm to share the details with you here.

We are building a SIC perturbation repository for both analysis and structural errors. Analysis errors are computed as differences between monthly individual ERA5 members and ERA5 ensemble mean over 1979-2020. Structural errors are computed as differences between OSTIA REP and ESA-CCIv2 monthly fields over 1982-2020. Hence we have *spatial maps* of samples of these errors. All fields are interpolated on a 1x1 grid and differences in the mean state removed.

The perturbation repository is stratified by calendar month and date range to capture uncertainties that correspond to the season and the level of sampling of the ocean by the observing system. For any given date, the perturbation patterns are randomly selected within the corresponding period of the repository and added to the corresponding field (i.e. done consistently for both sea ice and ocean perturbations).

I want to be clear: I completely agree that it is best to test the methodologies for distributing the sea ice into thickness categories in the control, unperturbed, deterministic simulation, and then validating the sea ice concentration fit against the assimilated observations. (I.e. I am not suggesting any changes to your methodology.) However, I believe results such as the degraded (Atlantic) sea ice edge thickness (Figure 4) when employing the gamma thickness distribution are likely related to uncertainties in the ice concentration observations – and this should probably be acknowledged. Similarly, the need to suppress temperature increments at the expense of sea ice increments (Section 6.3) is again likely a result of uncertainties in both the sea ice observations and SST analysis close to the sea ice edge as covered in my next point.

- Section 6.3: I have a small issue with referring to the suppression of temperature increments as a balance relationship. For a balance constraint, typically one would have a balanced temperature increment dependent on the sea ice, but then an additional unbalanced term and unbalanced increments to temperature that could still improve the fit to temperature observations. But more philosophically, here you are literally placing your finger on the balance scale by tipping the resulting analysis fit towards the sea ice observations and away from the temperature observations. If anything, it would be more akin to introducing bogus (zero innovation) observations in the presense of sea ice. All these methods, along with still others are employed to improve retention of increments in the analysis, so I have no issue with the method, just perhaps the terminology.

This is the main issue at hand, so we are very glad to see it is clear until this point. We are happy to replace the term "balance" with "physical relationship" to improve communication around this terminology.

However, my understanding is not complete, and maybe they are all interconnected. When first I looked at the results it is clear that the fit to sea ice does improve with increasing alpha – but it was not clear that your choice of alpha was actually maximizing that fit (i.e. would it be a good idea to test a higher value of alpha). With better understanding, I now believe there would be no maximization (although perhaps saturation); continuing to increase alpha should continue to increase the fit to the sea ice observations, although presumably in detriment to the fit to temperature observations and presumably with lowering levels of improvement as alpha is further increased. What is shown (Tables 4 &5; Figures 6,7 &8) is only half the story as presumably this is all at the cost of sea surface and sub-surface temperature statistics (but only in the vicinity of the sea ice edge). This then connects with item 1: How does this connect with uncertainties in the sea ice concentration and temperature observations? Are you improving a fit to observation (in reality, that "observation" is actually an analysis) to a point the fit is well below the significant uncertainty in the observation? Figure 7, 8 & 9 of Renfrew et al. [2021] would seem to suggest this might be so. Is there a need to re-address this issue with post 2025-07-06 results and the introduction of the OSI-438 (AMSR2) product (Table 1)?

This is an excellent summary which hopefully we can help expand on to make it clearer.

Please see our responses to RC1 and RC2 who both suggested increasing $\alpha$ further to partially answer your question here.

Ocean temperature observations are few and far between near the ice edge or under the ice pack, so when we look at observation fits to temperatures we do not see degradations (these are not shown as neutrality seemed not very interesting for the reader). So we could continue to increase $\alpha$ and not see degradations, all the while we are making larger and potentially unconstrained changes to the ocean below the ice. It might be that instead of the ocean temperatures being the reason for ice increments not being retained, it might be instead the atmospheric forcing that means ice increments are rejected.

Minor Comments

1. I think it should be pointed out other methods of distributing sea ice concentration amongst the thickness categories do exist. For example, Smith et al. [2016] propose the rescaled forecast tendencies (RFT) method that is largely identical to your method. For instance their Eqn. 1 is identical to your equation 3. Please cite.

Thank you very much for pointing this out. We were unaware of this work and it clearly needs to be cited and discussed. We think our method is the same as their Rescale the Existing ice thickness Distribution (RED) method and not their Rescaled Forecast Tendencies (RFT) method. "In effect means that the increment is distributed in proportion to the existing ice thickness distribution as in the *Rescale the Existing ice thickness Distribution (RED)* method of Smith et al. [2016]".

We will further add to the conclusions some discussion on their RFT methodology:

"We have shown that results are highly sensitive to the method used to distribute a single category increment across the multiple prognostic thickness categories of the model. Our choice to keep the increments in proportion to the background profile of ice distribution was shown to perform well, and has the benefit of leaving the sea ice thickness unaffected, by design. There are other methods in the literature for distributing ice increments across thickness categories, such as the Rescaled Forecast Tendencies (RFT) method of Smith et al. [2016]. Such methods that estimate what type of model deficiency led to an increment to the sea ice concentration, and apply concentration increments in such a way to counteract them. We leave comparisons against such methods for future investigations."

However reading their paper it is not clear which forecast lead time to use to appropriately compute such forecast tendencies given the First Guess at Appropriate Time (FGAT) methodology we use. We will need to discuss with Smith et al. in this regard.

2. Joint Minimization (l. 114): It has been my understanding that joint minimimization of the univariate sea ice and multivariate ocean has always been a problem that leads to degraded fits, so I am happy that you have managed to achieve this. However, will this not be a function of the observing system. Could improved numbers of near ice temperature observations change this dynamic? I suppose the hope is to go toward a more balanced, joint multi-variate ocean sea-ice assimilation before any substantial changes occur in the observing network.

That was our previous experience, when assimilating level 4 sea ice concentration products. It was implemented somewhat naively, sampling sea ice observations globally where clearly the data is of no use in regions such as the tropics. This led to an imbalance which observations the minimsation targetted with its gradient descent method. The two key developments were moving to L3 sea ice observations where missing data is accounted for appropriately and we do not sample regions far from sea ice, and the modifications to the oceanic component of the background error covariance matrix as documented in Chrust et al. [2024].

3. Figure 4 (Section 6.1). As stated above in Major Item #1, I believe the degradation in sea ice thickness when using the gamma distribution is ultimately due to the uncertainties in the sea ice concentration and location of the ice edge. Although we are given insufficient information to confirm this, I suspect the gamma distribution has a bias towards thicker ice, at least at the ice edge (what is the hemisphere change in bias?) – which then shows up as increased RMSE there as the thickness observations would be sensing zero thickness ice. (I.e. Your decision – I believe correct decision – to not utilize the slightly better in terms of ice concentration, gamma distribution, is likely due to offsetting biases.) This might be a result that could change in the perturbed solution. It also again would likely change post 2025-07-06 with the introduction of OSI-438. We agree. Hopefully the new figure we propose as a response to Reviewer 2's comments will also address this point - it is indeed a bias in the marginal ice zone which we see.

4. Open Water (Section 6.2) I could not help but think your open water fit results were one-sided. The results continue to get better as you increase the thickness ice increments in open water. While I would agree adding ice to open water into the 2nd category would seem weird – and would disconnect your data assimilation new ice thickness to your thermodynamic new ice thickness. (Actually that wording is a little ambigouous in Section 4.3.2 – the new ice thickness in the DA is or is not identical to the thermodynamic new (frazil) ice – or just the other properties?) At any rate, the same arguments that the data assimilation process is correcting for errors in dynamical movement of ice as much as errors in thermodynamics as justification for adding ice across all categories could presumably be used to justify adding higher category (relocated ice) for positive increments in open ice. Sorry: I seem to recall a statement is made along these lines, but I could not track it down in the manuscript. It might have been interesting to test adding 2nd category thickness ice to see where the fit to observations begins to decrease.

We understand this point of view. It has certainly been a technical limitation that stopped us from increasing the thickness of new ice to open water beyond the maximum allowed in the thinnest category. As for the choices we made to distribute increments to the existing ice pack, we are very much hoping that (passive microwave) observations of sea ice thickness will ultimately tell us what we should be doing in these areas.

Until that point we want to keep the system as simple as possible. Once we start adding thicker ice, would we add distribute across the first 2 ice thickness categories? Or put it all in the second? These would be interesting to investigate, but will have to be left for future work.

"The other sea ice properties are set so that the sea ice created by the DA is identical to sea ice that would have thermodynamically formed into the open water." we should reword this to read

"The properties of this new sea ice are the same as frazil new ice formed themodynamically by the model from open water."

5. Table 4&5/Figure 8: If I have not missed something, $\alpha$ (Eqn. 5) has units of °C or K (it is a change in temperature – so your choice). The values of $\alpha$ in Tables 4&5 and Figure 8 – and in the text if they occur there – should be accompanied with that unit.

Happy to add that.

6. I will forego my anonymity as penance for my tardiness in achieving this review

Tardiness is welcome in this case, and we greatly appreciate your time and effort to help us improve this paper.